# Compositional Generalization for Data-to-Text Generation

**Xinnuo Xu**[1], **Ivan Titov**[1,2] and **Mirella Lapata**[1]
[1]ILCC, School of Informatics, University of Edinburgh
[2]ILLC, University of Amsterdam
xxu3@ed.ac.uk, {ititov, mlap}@inf.ed.ac.uk

## Abstract

Data-to-text generation involves transforming structured data, often represented as predicate-argument tuples, into coherent textual descriptions. Despite recent advances, systems still struggle when confronted with unseen combinations of predicates, producing unfaithful descriptions (e.g., hallucinations or omissions). We refer to this issue as compositional generalisation, and it encouraged us to create a benchmark for assessing the performance of different approaches on this specific problem. Furthermore, we propose a novel model that addresses compositional generalization by clustering predicates into groups. Our model generates text in a sentence-by-sentence manner, relying on one cluster of predicates at a time. This approach significantly outperforms T5 baselines across all evaluation metrics. Notably, it achieved a 31% improvement over T5 in terms of a metric focused on maintaining faithfulness to the input.[1]

## 1 Introduction

Data-to-text generation (DTG) (Gardent et al., 2017; Dušek et al., 2020) aims to accurately generate textual descriptions from input tuples; the tuples should encompass all the information needed for generating the description regardless of the narrative order. Typically, as shown in Figure 1, each tuple consists of two arguments and one predicate that conveys their relationship.[2] Given the large number of pre-defined predicates, it is time-consuming to collect human-annotated training examples for each potential combination of them. Thus, models must have the ability to generalize and handle examples with previously-unseen predicate combinations. We refer to this generalization scenario as *compositional generalization*.

Prior research (Mehta et al., 2022; Xu et al., 2021; Kale and Rastogi, 2020a; Peng et al., 2020;

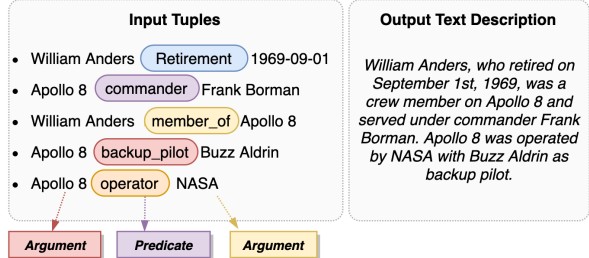

Figure 1: An example from WebNLG dataset (Gardent et al., 2017). DTG aims at transforming the input structured data (left) into coherent textual description (right).

Chen et al., 2020) has focused on evaluating the compositional generalization (CG) abilities of DTG models. These studies created few-shot training splits using established benchmarks by reducing the number of training examples or limiting the number of distinct predicate combinations in the training set through random selection. However, these arbitrary selections overlook the practical effort required for annotating different examples. For example, annotating examples with a larger number of input tuples requires more time and effort.

We introduce a test environment based on Gardent et al. (2017). During training, models are exposed to examples with fewer input tuples, while in the testing phase, examples with more input tuples are presented. To make it even more challenging, we combine CG with *few-shot learning* by reducing the number of training examples for each predicate combination to one. We also incorporate CG with *domain adaptation* by evaluating the models on unseen domains. Our results demonstrate that the SoTA pre-trained language models (LMs; Raffel et al. 2020; Kale and Rastogi 2020b) fail to generalize effectively in our experimental setup.

To tackle this issue, we propose a clustering-based method (Figure 2) that utilize the graph weights learned from training data to decompose unfamiliar predicate compositions into smaller groups during inference. Each group consists of predicate combinations encountered by the model

---

[1]Code available at: github.com/XinnuoXu/CG_DTG.
[2]We follow the format specified in Gardent et al. (2017).

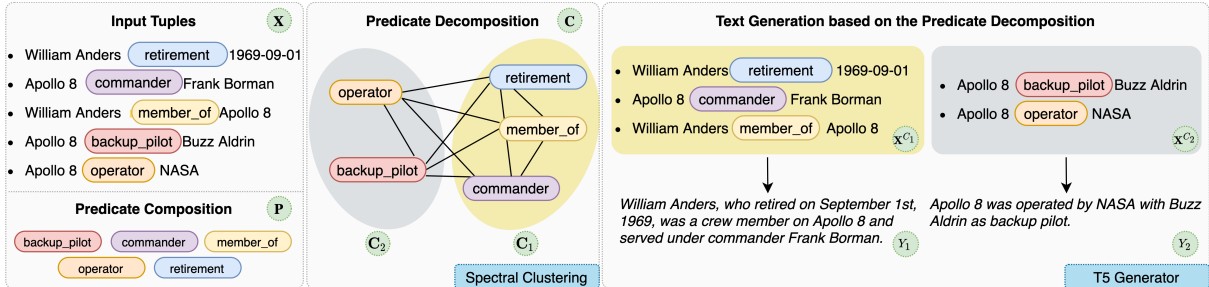

Figure 2: Framework of our proposed inference procedure. We introduce a set of clustering-based methods that leverage the graph weights learned during training to decompose the unseen predicate compositions into familiar groups. For each group, we gather the input tuples associated with predicates in that group, and generate a sentence to describe them. The final text description is created by combining the generated sentences from all the groups.

during training. Then, individual sentence descriptions are generated separately for each group, and combined to form the final description for the input. In contrast to previous studies that primarily rely on self-training to improve CG in DTG (He et al., 2020; Heidari et al., 2021; Li et al., 2021; Mehta et al., 2022), as well as using data augmentation to improve CG in various tasks such as semantic parsing (Andreas, 2020; Qiu et al., 2022; Fang et al., 2023), our method solely relies on small training sets and does not require any additional human-annotated, automatically labeled, or unlabeled data. In the CG-centric testing scenario, we observe significant improvements across all metrics in the benchmark compared to the vanilla T5 model (Raffel et al., 2020; Kale and Rastogi, 2020b). A faithfulness-based metric (Dušek and Kasner, 2020) shows an impressive gain of 31% over T5. A similar trend is seen when combining the CG challenge with few-shot learning and domain adaptation. Our contributions are:

• We create a benchmark to assess the CG ability of DTG models and generate four testing scenarios of varying difficulty by combining CG with few-shot learning and domain adaptation.

• We present an innovative architecture that uses a clustering algorithm to decompose the text description generation for unfamiliar input predicate combinations into smaller, familiar ones.

• We show that our method produces outputs that are not only more faithful but also exhibit a greater resemblance to human-written references compared to vanilla pre-trained LMs while tested on the proposed benchmark.

• We also introduce an intrinsic evaluation framework for inspecting input decomposition.

|  | Train on CGFull-k | Train on CGOneShot-k |
|---|---|---|
| Seen test | Compositional Generalization [1] | Compositional Generalization [2] Few-shot Learning |
| Unseen test | Compositional Generalization [3] Domain Adaptation | Compositional Generalization [4] Domain Adaptation Few-shot Learning |

Figure 3: Four evaluation scenarios.

## 2 CG focused Benchmark

Existing benchmarks (Ratnaparkhi, 2000; Liang et al., 2009; Mairesse et al., 2010; Banik et al., 2013; Wen et al., 2015; Lebret et al., 2016; Wen et al., 2016; Gardent et al., 2017; Wiseman et al., 2017; Novikova et al., 2017; Parikh et al., 2020) have provided an important test-bed for DTG models. We specifically choose WebNLG 2017[3] (Gardent et al., 2017) to build our benchmark upon (reasons are shown in Appendix B) and primarily focus on assessing the models' capability for CG. We present two sets of training splits and create four distinct testing scenarios with different difficulty levels by incorporating the training splits with the *seen* and *unseen* test sets offered in WebNLG.

### 2.1 Training Sets

**CGFULL** consists of a set of independent training splits {CGFULL-k}, k ranging from 2 to 7. CGFULL-k exclusively consists of examples where the number of input tuples is equal to or less than k. We excluded CGFULL-5 and -6 here due to the marginal increase in the amount of data within these two sets. Note that, CGFULL-7 represents the full training set of WebNLG 2017.

---

[3] https://github.com/ThiagoCF05/webnlg/tree/master/data/v1.6.

| Training Splits | CGOneShot | | | | | | CGFull | | | |
|---|---|---|---|---|---|---|---|---|---|---|
| | -2 | -3 | -4 | -5 | -6 | -7 | -2 | -3 | -4 | -7 |
| # Examples | 553 | 987 | 1477 | 1867 | 2043 | 2203 | 7712 | 11350 | 14744 | 18101 |
| % of WebNLG2017 train | 3% | 5% | 8% | 10% | 11% | 12% | 43% | 63% | 81% | 100% |

Table 1: The amount of examples within splits in collection CGFull and CGOneShot.

**CGOneShot** is a one-shot version of CGFull. We define *predicate composition* as a combination of the input predicates that is order agnostic. In CGFull, the same *predicate composition* is found in multiple training examples paired with different arguments. To present a more challenging situation, we generate split CGOneShot-k by randomly selecting one example for each *predicate composition* in the corresponding CGFull-k split.

The statistical details are presented in Table 1. Models will undergo separate training on each individual splits in both CGFull and CGOneShot.

## 2.2 Validation and Test Sets

The validation and test set in the original WebNLG 2017 remain unchanged. The test set consists of two categories: The *seen* category includes examples from 9 domains present in training, while the *unseen* category consists of examples from 5 new domains with newly defined predicates and unseen entities not present in training. Note that, both validation and test sets contain examples with different numbers of input tuples, ranging from 1 to 7.

## 2.3 Evaluation Scenarios

We create four evaluation scenarios (Figure 3) by pairing training splits with test sets. When trained on CGFull-k, the models are exposed to numerous examples with predicate combinations of up to $k$ predicates. When tested on the WebNLG *seen* category, their CG abilities are evaluated as they encounter novel combinations consisting of a greater number of predicates. To further intensify the challenge, the models are tested on the *unseen* category, requiring them to demonstrate both CG and adaptability to predicate combinations from new domains. They also need to handle newly introduced arguments. On the other hand, the models trained on CGOneShot-k have only seen one example for each combination of up to $k$ predicates. When tested on the *seen* set, their CG abilities are assessed with few-shot learning skills. When evaluated on the *unseen* set, the examination of their capabilities further extends to domain adaptation.

## 2.4 Evaluation Metrics

**Generation Evaluation** focuses on evaluating the generated text w.r.t. its similarity to human-authored reference sentences. We adopt BLEU (Papineni et al., 2002), a token-level exact matching metric that is incorporated in the WebNLG.

**Faithfulness Evaluation** tests if the generated text is faithful to the input tuples (Wen et al., 2015; Reed et al., 2018; Dušek and Kasner, 2020). Unfaithful generations contain hallucinations (generations with extra or incorrect information) or omissions (generations missing important input information) (Dušek et al., 2019). We adopt PARENT (Dhingra et al., 2019), an entailment-based metric, where a higher score indicates a lower occurrence of hallucinations; and OK-percent (Dušek and Kasner, 2020), a natural language inference-based metric representing the proportion of system generations free from hallucinations or omissions.

## 3 Clustering based CG Methods

The framework of our approach is shown in Figure 2. For clarity, we denote the input tuples and their corresponding predicates as $\mathbf{X} = \{X_1, X_2, \cdots X_N\}$ and $\mathbf{P} = \{P_1, P_2, \cdots P_N\}$ respectively, where $P_i$ is the predicate of the $i^{th}$ tuple $X_i$. $N$ represents the total number of tuples. The output text is denoted as $\mathbf{Y} = \{Y_1, Y_2, \cdots Y_M\}$, where $Y_j$ represents the $j^{th}$ sentence in the output.

Fine-tuning pre-trained LMs (Kale and Rastogi, 2020b; Ribeiro et al., 2021), aim to maximize the log likelihood of generating the ground truth text $\mathbf{Y}$ given the linearized input tuples $\mathbf{X}$, denoted as $\log p(\mathbf{Y}|\mathbf{X})$, during training. However, these models face challenges in generalizing to unseen predicate combinations. To overcome this, an intuitive approach is to decompose these unseen combinations into smaller groups, ensuring that the combination in each group has been seen during training; then, generate a sentence from each group individually and combine them to form the final description.

We denote the decomposition as $\mathbf{C} = \{\mathbf{C}_1, \mathbf{C}_2, \cdots \mathbf{C}_M\}$, where $\mathbf{C}_j$ represents the $j^{th}$ predicate group responsible for generating sentence $Y_j$. Since DTG tasks require to include all the input

information in the output without repetition, the decomposition must fulfill $\mathbf{C}_1 \cup \mathbf{C}_2, \cdots \cup \mathbf{C}_M = \mathbf{P}$ and $\forall i, j : 1 \leqslant i < j \leqslant M$, $\mathbf{C}_i \cap \mathbf{C}_j = \varnothing$. The text generation is then broken down into a set of parallel steps. Each step aims at creating a single sentence to describe the tuples associated with the predicates in one of the groups:

$$p\left(\mathbf{Y}|\mathbf{C}, \mathbf{X}\right) = \prod_{j=1}^{M} p\left(Y_j|\mathbf{X}^{C_j}\right) \tag{1}$$

where $\mathbf{X}^{C_j}$ is a subset of $\mathbf{X}$. The tuple $X_i \in \mathbf{X}^{C_j}$ iff its predicate $P_i \in \mathbf{C}_j$.[4] An alternative representation of a predicate decomposition involves the use of a matrix. Given a set of tuples with their corresponding predicates, we construct a fully connected undirected graph, denoted as $G = (V, E)$, where the predicates are represented as nodes in $V$. In turn, a decomposition can be considered as a partitioned graph derived from the original graph $G$. We encode the partitioned graph as a binary matrix $\mathbf{M}$, where $\mathbf{M}_{ij} = 1$ signifies that the predicates $P_i$ and $P_j$ belong to the same group, while $\mathbf{M}_{ij} = 0$ indicates that they belong to different groups.

Unfortunately, the annotated ground truth decompositions are unavailable in the majority of DTG training sets. Therefore, the training objective becomes maximizing the marginal log likelihood of the output text $\mathbf{Y}$ w.r.t. the latent $\mathbf{M}$.

$$\log p\left(\mathbf{Y}|\mathbf{X}\right) = \log \mathbb{E}_{\mathbf{M} \sim p(\mathbf{M}|\mathbf{X})} p\left(\mathbf{Y}|\mathbf{M}, \mathbf{X}\right) \tag{2}$$

As the number of input tuples increases, exploring all possible decompositions for each example is intractable. Following Kim et al. (2017); Deng et al. (2018), we approximate the marginal likelihood as:

$$\log \mathbb{E}_{\mathbf{M} \sim p(\mathbf{M}|\mathbf{X})} p\left(\mathbf{Y}|\mathbf{M}, \mathbf{X}\right) \approx \log p\left(\mathbf{Y}|\mathbb{E}_{\mathbf{M} \sim p(\mathbf{M}|\mathbf{X})}\mathbf{M}, \mathbf{X}\right)$$

This results in the stochastic decomposition variable $\mathbf{M}$ being replaced with the deterministic value $\bar{\mathbf{M}} = \mathbb{E}_{\mathbf{M} \sim p(\mathbf{M}|\mathbf{X})}\mathbf{M}$. Assuming that $\mathbf{M}$ follows a Bernoulli distribution $\mathbf{B}(\gamma)$, with each element within the matrix being independent, we can represent the distribution of $\mathbf{M}$ as $\mathbf{M}_{ij} \sim \mathbf{B}(\gamma_{ij})$. Thus, the expectation of the binary matrix $\bar{\mathbf{M}}$ is the Bernoulli parameter matrix $\gamma$. In Section 3.1 we demonstrate two training methods to predict $\gamma$.

## 3.1 Training

Inspired by Su et al. (2021) and Moryossef et al. (2019), we propose an automatic way for creating

silver annotations for the training of $\gamma$. For each training example in DTG[5], we calculate the BLEU score for each input tuple w.r.t. each sentence in the reference output. Afterwards, the tuple is aligned with the sentence that achieves the highest BLEU score. This process yields a collection of tuple groups, where tuples within a group are described in the same sentence. By removing the arguments from each tuple, we obtain an annotated predicate decomposition from each DTG training example. Now, we introduce two methods to obtain matrix $\gamma$ using these annotated predicate decompositions:

**Numerical Weight Prediction** determines $\gamma_{ij}$ by analyzing individual occurrence vs. co-occurrence of two predicates $P_i$, $P_j$ in the training data:

$$\gamma_{ij} = \frac{\#\left(P_i, P_j\right)}{\#\left(P_i\right) + \#\left(P_j\right)}$$

where $\#\left(P_i, P_j\right)$ denotes the frequency of both predicate $P_i$ and $P_j$ being mentioned in the same sentence throughout the corpus. $\#\left(P_*\right)$ represents the frequency of predicate $P_*$ appearing in the corpus[6]. However, this approach has a limitation: if either predicate is not included in the training set, the weight will always be zero. This becomes challenging when transitioning to a new domain, since most weights in the matrix $\gamma$ will be zero.

**Neural Network based Prediction** solves this problem by introducing a small scale transformers-based neural network[7]. The model takes two tokenized predicates concatenated as input, including a "[CLS]" token attached to the beginning. The embedding of the "[CLS]" token from the final transformer layer is then fed into a classification head. This head predicts whether the two predicates should be described in the same sentence (1) or not (0). Elements in $\gamma$ can be written as:

$$\gamma_{ij} = sigmoid(\mathbf{W}\left(\text{Transformers}\left(P_i, P_j\right)\right) + \mathbf{b})$$

where $\mathbf{W}$ is a linear transformation. $\mathbf{b}$ is the bias. For classifier training, synthetic data is generated using the automatically annotated predicate decomposition. Positive examples are formed by pairing any two predicates within the same group, while negative examples consist of pairs from different groups. In cases where there is only one predicate

---

[4]For the input tuple subsets, we also have $\mathbf{X}^{C_1} \cup \mathbf{X}^{C_2}, \cdots \cup \mathbf{X}^{C_M} = \mathbf{X}$ and $\forall i, j : 1 \leqslant i < j \leqslant M$, $\mathbf{X}^{C_i} \cap \mathbf{X}^{C_j} = \varnothing$.

[5]Check Appendix C for the data preprocessing details.

[6]To calculate $\#\left(P_*\right)$, we discard the examples with only a single tuple as input.

[7]Appendix D shows the hyper-parameters.

---

**Algorithm 1** DTG with Predicate Decompostion

---
**Require:** Input tuples $\mathbf{X}$; Their predicates $\mathbf{P}$; Trained predicate clustering model $\mathbf{PC}$; Fine-tuned generator $\mathbf{T5}$.
 1: **for** $k \leftarrow 1$ to $|\mathbf{X}|$ **do**
 2:     $\mathbf{C} \leftarrow \mathbf{PC}(\mathbf{P}, k)$
 3:     **if** EffectiveCluster($\mathbf{C}$) **then**
 4:         $\bar{\mathbf{C}} \leftarrow \mathbf{C}$ and $\bar{k} \leftarrow k$; Break
 5:     **end if**
 6: **end for**
 7: $\bar{\mathbf{C}} \leftarrow$ Sort($\bar{\mathbf{C}}$)
 8: **for** $j \leftarrow 1$ to $\bar{k}$ **do**
 9:     $Y_j \leftarrow \mathbf{T5}(\{\mathbf{X}^{\bar{C}_j}\})$
10: **end for**
11: $\mathbf{Y} \leftarrow \{Y_j\}$

---

group in the input, indicating a single-sentence output, each input predicate is randomly paired with a predicate from the dataset that is not part of the input for the negative examples' creation. The model is trained with the Cross Entropy loss.

To train the text generation model, we use the annotated tuple groups introduced earlier in this section instead of relying on the predicate decompositions predicted by the introduced models. Since each tuple group is aligned to a sentence, we take the tuples as input and the sentence as output to fine-tune a setence-level T5 [8] for text generation.

### 3.2 Testing

The testing procedure is outlined in Algorithm 1. Initially, we aim to obtain a predicate decomposition $\bar{\mathbf{C}}$ for a given set of input tuples. To accomplish this, we begin by estimating the expectation of the binary matrix $\mathbf{M}$, i.e. $\gamma$, using the models introduced in Section 3.1. Afterwards, we iterate through all possible values for the number of predicate groups $k$ (ranging from 1 to $|\mathbf{X}|$) and employ a clustering algorithm (specifically, spectral clustering in this study) over the matrix $\gamma$ [9]. If the minimum weight between two predicates within the same cluster exceeds a threshold $\epsilon$ [10], we halt the exploration. Our objective is to minimize the number of predicate clusters and ensure that each cluster does not contain unfamiliar predicate pairs.

To enhance the coherence of the generated text, we implement a simple method to arrange the pred-

---

[8]In our experiments, we observed a decrease in the BLEU metric when using sentence-level T5 models for text generation. Consequently, we substitute them with standard tuples-to-paragraph T5 models that are fine-tuned on unaligned input-output data in each training split.

[9]To improve the coherence of the generated texts, we adjust $\gamma$ by assigning 0 to $\gamma_{ij}$ if the corresponding triples of predicate $P_i$ and $P_j$ do not share any arguments.

[10]The threshold $\epsilon$ is determined using the validation set.

---

icate clusters. For each cluster, we calculate the occurrence of its predicates being described in the first sentence across all training examples, and choose the one with the highest frequency as the first cluster. Next, we select the subsequent cluster by identifying the one with the highest number of unique arguments observed in the previous cluster. We repeat this step until all clusters are sorted. Finally, we utilize the fine-tuned T5 model to produce a sentence for each cluster following the order, and concatenate these generated sentences to form the final output describing the input tuples.

### 3.3 REINFORCE-enhanced Decomposition

Deterministic approaches heavily rely on automatically annotated predicate decompositions. However, the annotator based on exact token matching is weak at detecting paraphrasing, resulting in misalignment of tuples to the wrong sentence. To address this, we propose a REINFORCE (Glynn, 1990; Williams, 1992) based approach that reduces reliance on silver annotations.

We first simplify the marginal distribution in Eq 2 using Jensen's Inequality. Since the logarithm function is concave, we have:

$$\log \mathbb{E}_{\mathbf{M} \sim p(\mathbf{M}|\mathbf{X})} p(\mathbf{Y}|\mathbf{M}, \mathbf{X}) \geqslant \mathbb{E}_{\mathbf{M} \sim p(\mathbf{M}|\mathbf{X})} \log p(\mathbf{Y}|\mathbf{M}, \mathbf{X})$$

Our goal is to train the parameters $\phi$ in $p_\phi(\mathbf{M}|\mathbf{X})$ to optimize this lower bound. The gradient of $\phi$ is:

$$\mathbb{E}_{\mathbf{M} \sim p_\phi(\mathbf{M}|\mathbf{X})} [\log p(\mathbf{Y}|\mathbf{M}, \mathbf{X}) \bigtriangledown_\phi \log p_\phi(\mathbf{M}|\mathbf{X})]$$

However, directly sampling a binary matrix from the Bernoulli distribution $\mathbf{M} \sim \mathbf{B}(\gamma)$ does not guarantee that the matrix can be transformed into a valid set of clusters $\mathbf{C}$. Therefore, we propose a method to replace the sampling process in the forward pass. We first sample a binary matrix $\mathbf{M}$ from the Bernoulli distribution $\mathbf{B}(\gamma)$. Next, we perform element-wise multiplication between this matrix and $\gamma$, denoted as $\mathbf{M} \odot \gamma$. Finally, we apply the spectral clustering algorithm to the resulting matrix to obtain discrete clusters $\mathbf{C}$. [11] As this process is not differentiable, akin to the concept of the straight-through estimator (Bengio et al., 2013), we perform a backward pass through the sampling step by computing the probability $p_\phi(\mathbf{M}|\mathbf{X})$ using the sampled predicate decomposition $\mathbf{C}$:

$$p_\phi(\mathbf{M}|\mathbf{X}) = \prod_{i=1}^{N} \prod_{i=1}^{N} \gamma_{ij}^{k_{ij}} (1 - \gamma_{ij})^{1 - k_{ij}}$$

---

[11]The number of clusters corresponds to the number of sentences in the ground truth text description.

where $k_{ij} \in \{0, 1\}$. $k_{ij} = 1$ means that predicate $P_i$ and $P_j$ belong to the same cluster in $\mathbf{C}$, while $k_{ij} = 0$ indicates that they are assigned to different clusters. By using this approach, the gradients can be propagated through the Bernoulli distribution. We compute $\gamma_{ij} \in (0, 1)$ using the transformers classifier structure discussed in Section 3.1. In order to speed up convergence, we initialize the parameters using the classifier that has been trained in the deterministic approach.

In turn, we align each cluster in the sampled decompositon $\mathbf{C}$ with a sentence from the ground truth text description utilizing the Hungarian algorithm (Kuhn, 1955). The cost matrix for the algorithm is derived from the negative BLEU score between a tuple group and a sentence. Then, we employ the fine-tuned T5 generator (Section 3.1) as the reward model to evaluate the sampled predicate decomposition. The reward is calculated based on Eq 1, which involves multiplying the likelihood of generating each sentence in the ground truth, conditioned on its corresponding aligned tuple group. Since REINFORCE is prone to high variance (Zaremba and Sutskever, 2015; Li et al., 2016), we propose a baseline $\log p(\mathbf{Y}|\tilde{\mathbf{C}}, \mathbf{X})$, to the reward model. $\tilde{\mathbf{C}}$ denotes a randomly generated predicate decomposition. We achieve this by randomly assigning each tuple to a cluster.[12]

## 4 Experiments and Results

We compare our methods (CG-Numerical, CG-NN, CG-RL) against fine-tuning T5[13] (Kale and Rastogi, 2020b) using the benchmarks introduced in Section 2. Additionally, we include another baseline called CG-Random. It determines the number of groups, ranging from 1 to $\mathbf{X}$ and assigns each predicate to one of the groups randomly. Comparing to CG-Random allows us to gain insights into the impact of our methods on predicate decomposition and how it affects the text generation quality. To enhance readability given the large number of experiments conducted, we present a portion of the results that will be thoroughly discussed in this section. Table 2 and 3 show models performance in testing scenario 2 and 4 (refer to Figure 3), respectively. For scenario 1 and 3, the results can be

---

[12]If any clusters end up without tuples, we repeat the process until all clusters contain at least one tuple.

[13]Mehta et al. (2022) found that simply increasing the model size does not effectively bridge the CG gap. Due to resource constraints, we specifically focus on T5-base. Hyperparameters for fine-tuning is shown in the released code.

| | | CGOneShot | | | | | |
| | | -2 | -3 | -4 | -5 | -6 | -7 |
|---|---|---|---|---|---|---|---|
| T5 | BLEU | 44.94 | 49.43 | 54.39 | 56.86 | **58.49** | **58.98** |
| CG-Ra | | 47.01 | 49.03 | 49.32 | 51.06 | 50.39 | 51.08 |
| CG-Nu | | 47.46 | 52.91 | 55.37 | **57.81** | 56.32 | 58.07 |
| CG-RL | | **47.58** | **53.17** | **57.06** | 57.41 | 58.08 | 58.44 |
| T5 | PARENT | 46.03 | 50.61 | 53.75 | 54.33 | 55.14 | 56.12 |
| CG-Ra | | 50.43 | 53.14 | **55.37** | 55.39 | 56.04 | 56.75 |
| CG-Nu | | 52.23 | **53.26** | 55.24 | 55.39 | 55.90 | **56.92** |
| CG-RL | | **52.11** | 52.84 | 54.33 | 55.24 | 55.47 | 56.52 |
| T5 | OK-per | 36.56 | 47.79 | 64.16 | 68.07 | 74.46 | 78.37 |
| CG-Ra | | 59.22 | **68.38** | **77.34** | **79.09** | **81.26** | **81.15** |
| CG-Nu | | **64.88** | 62.92 | 71.68 | 73.02 | _79.92_ | _80.23_ |
| CG-RL | | _63.65_ | 61.28 | 68.92 | _74.67_ | 77.24 | 79.40 |

Table 2: Performance of models evaluated in scenario 2 (refer to Figure 3), i.e. trained on **CGOneShot**-k and tested on *SEEN* category. The top-performing system is highlighted in bold, while the second best system for Ok-percent is underlined. CG-Ra, CG-Nu are short for CG-Random and CG-Numerical, respectively.

| | | CGOneShot | | | | | |
| | | -2 | -3 | -4 | -5 | -6 | -7 |
|---|---|---|---|---|---|---|---|
| T5 | BLEU | 35.80 | 39.79 | **44.57** | **45.06** | **47.43** | **47.69** |
| CG-Ra | | **39.19** | **40.00** | 40.57 | 41.51 | 42.46 | 41.87 |
| CG-Nu | | 35.31 | 36.83 | 37.66 | 38.21 | 39.83 | 39.13 |
| CG-RL | | 36.06 | 37.94 | 39.86 | 40.73 | 42.38 | 42.84 |
| T5 | PARENT | 37.79 | 40.18 | 44.96 | 46.59 | 47.86 | 48.14 |
| CG-Ra | | 41.90 | 43.41 | 46.19 | 48.25 | 48.30 | 48.12 |
| CG-Nu | | 44.48 | **44.15** | **46.78** | 48.17 | 48.82 | 48.58 |
| CG-RL | | **44.67** | 43.87 | 46.64 | **48.72** | **49.22** | **49.19** |
| T5 | OK-per | 34.34 | 41.19 | 53.42 | 55.22 | 57.46 | 64.31 |
| CG-Ra | | 52.41 | 55.33 | 63.86 | 64.76 | 66.55 | 70.15 |
| CG-Nu | | **70.82** | **69.02** | **71.38** | **71.49** | **71.60** | **72.62** |
| CG-RL | | _67.90_ | _68.01_ | _68.24_ | _71.16_ | _69.25_ | _72.50_ |

Table 3: Performance of models evaluated in scenario 4 (refer to Figure 3), i.e. trained on **CGOneShot**-k and tested on *UNSEEN* category. CG-Ra, CG-Nu are short for CG-Random and CG-Numerical, respectively.

found in Table 10 and 11 in Appendix F.

### 4.1 Case Study on Pre-trained LMs

In all four testing scenarios, we observe a significant decline in both generation performance and faithfulness when T5 is trained on the splits with fewer input tuples. This suggests that pre-trained LMs are not well-suited for CG tasks. Additionally, we highlight the performance of T5 models trained using CGOneShot-7, CGFull-2 and -7 in Table 4. Comparing CGFull-7 and CGOneShot-7, we observe that limiting the number of training examples for each predicate combination to one does not significantly hurt the performance. However, when comparing CGFull-7 with CGFull-2, a significant decrease in performance is observed. This drop occurs even though there is a smaller reduction in training data. These findings highlight the difficulty of our benchmark and emphasizes the

| T5 | | ONESHOT-7 | FULL-2 | FULL-7 |
|---|---|---|---|---|
| #Example | | 2203 | 7712 | 18101 |
| BLEU | *Seen* | 58.98 | 52.54 | 65.01 |
| PARENT | | 56.12 | 54.31 | 62.17 |
| OK-per | | 78.37 | 43.87 | 78.27 |
| BLEU | *Unseen* | 47.69 | 44.94 | 58.98 |
| PARENT | | 48.14 | 46.03 | 56.12 |
| OK-per | | 64.31 | 36.56 | 78.37 |

Table 4: Performance of T5 models trained on CGONESHOT-k (short as ONESHOT) and CGFULL-k (short as FULL). # Example represents the number of training examples in each training splits.

importance of models possessing CG abilities.

## 4.2 Results of Proposed Approaches

Our objective is to identify methods that excel across **all of the four** distinct testing scenarios introduced in Section 2.3.

**Our approaches *vs.* T5** Our approaches demonstrate superior performance compared to T5 in all four testing scenarios (refer to Figure 3), measured across all three metrics, except for BLEU in the domain adaptation scenario (Table 3). This is due to the presence of new predicates in the out-of-domain test set. Our approaches tend to decompose unseen predicate compositions into smaller groups for text generation, which deviates from human annotations. On average, the number of sentences in the descriptions generated by humans, T5, and our approaches are 1.35, 1.4, and 2.0 respectively (see Table 16 in Appendix I). This divergence is penalized when evaluating with reference-based metrics like BLEU. However, our study encourages the decomposition behavior as it allows for the generation of faithful texts while maintaining a reasonable level of similarity to the human-written references. Another observation is that, across all four scenarios, when trained solely on examples with fewer input tuples, the performance advantage of our approaches over T5 becomes more pronounced. For example, when trained with CGFULL-2 and tested on the *seen* set (Table 10 in Appendix F), the best performed CG-based approach outperform T5 2 points on BLEU, 4.2 points on PARENT and 31.2 points on OK-percent. These findings highlight the effectiveness of our approaches in enhancing the CG capability of vanilla pre-trained LMs, particularly when trained on a very limited number of examples with simple predicate compositions.

**Our approaches *vs.* CG-random** Our approaches generally outperform CG-random, particularly in terms of BLEU scores. However, there is a noticeable decrease in the OK-percent scores when tested using the *seen* set (Table 2). This is because predicate compositions in the in-domain test set are more likely to have been seen in the training set. Thus, our models tend to decompose input predicates into fewer groups. However, CG-random selects the number of groups randomly, resulting in a higher average group number (Table 16 in Appendix I). This allows CG-random to achieve slight gains in OK-percent but comes at the cost of an 8-point decrease in BLEU compared to our approaches. Our study does not promote this unnecessary decompositions that may lead to unnatural text descriptions. These findings indicate that breaking down predicate compositions into smaller groups generally results in more faithful generations. However, when compared to the random approach, learned decompositions produce texts that are closer to human expressions.

**CG-Numerical *vs.* CG-RL** In this section, we directly compare CG-Numerical to CG-RL since CG-RL is an extension of CG-NN. The comparison between CG-NN and CG-RL can be found in Appendix F. CG-RL exhibits better performance than CG-Numerical in terms of BLEU score across all scenarios, particularly when evaluated on out-of-domain examples (Table 3 and 11). The results for metrics PARENT and OK-percent are comparable between the two approaches, except that CG-Numerical consistently outperforms CG-RL in terms of OK-percent when tested in unseen domains (Table 3). The reason for this is that CG-RL utilizes neural networks to encode tokenized predicates, enabling some level of knowledge transfer when encountering out-of-domain predicates that consist of tokens seen in the in-domain predicates. However, CG-Numerical is unable to process out-of-domain predicates, resulting in a higher number of decomposed clusters. In fact, this number (2.4) is even higher than that of CG-random (1.8) (see Table 16 in Appendix I). Consequently, this contributes to a decrease in BLEU.

## 5 In-depth Analysis and Discussion

### 5.1 Qualitative Evaluation

For the qualitative evaluation, we randomly selected 30 inputs from the *seen* test set, each containing five or more tuples. All models were trained using CGFULL-2 and tasked with generating text descriptions for these inputs. Building upon the metrics used in prior works (Mehta et al., 2022;

| CGFULL-2 | Gr | Re | Ha | Om |
|---|---|---|---|---|
| T5 | 2.0 | 2.0 | 1.8 | 0.13 |
| CG-Random | 2.0 | 1.17 | 1.8 | 1.17 |
| CG-RL | 2.0 | 1.1 | 1.93 | 1.67 |

Table 5: Human evaluation with the metrics of Grammar, Repetition, Hallucination, and Omission. A higher score indicates better performance. Models are trained using **CGFULL**-k and tested on the *SEEN* set.

| CGFULL | -2 | -3 | -4 | -7 |
|---|---|---|---|---|
| CG-Random | 0.4087 | 0.3859 | 0.4355 | 0.3602 |
| CG-Numerical | 0.4487 | 0.5235 | 0.5703 | 0.6064 |
| CG-RL | **0.4755** | **0.6217** | **0.6251** | **0.6433** |
| Human | | 0.7006 | | |

Table 6: Predicate decomposition performance evaluated using NMI. Models are tested under scenario 1, i.e. trained using **CGFULL**-k and tested on the *SEEN* set.

Chen et al., 2020), we introduced four metrics: grammar (Gr), repetition (Re), hallucination (Ha), and omission (Om). Each system was rated on a scale of 0 (bad), 1 (moderate), and 2 (good) for each metric. Detailed evaluation standards can be found in Appendix E. The results are presented in Table 5. CG-RL outperforms T5 and CG-Random in hallucination and omission, showcasing its ability to generalize from 2-tuple examples to multiple tuples with accurate and faithful descriptions. In contrast, the generated texts from CG-RL exhibit more repetition compared to T5. One prominent issue observed in CG-RL is the repetitive sentence structure, such as generating multiple sentences that start with the same entity. On the other hand, the majority of texts generated by T5 are single-sentence, indicating that it suffer less from the repetition problem. In principle, we have the option to reintroduce conditional relationships between the generated sentences to mitigate the repetition of entities in the proposed CG-RL. Cherry-picked examples are shown in Table 8. Randomly-picked examples can be found in 9 in Appendix A.

## 5.2 Predicate Decomposition Performance

The WebNLG 2017 v1.6 test sets include human-annotated decompositions. For each input example, we randomly select two annotated decompositions from its references, ensuring they have an equal number of predicate clusters. One of them is chosen as the reference, while the other becomes the hypothesis. Examples with only one predicate cluster as output are excluded, resulting in a total of 160 testing examples. We evaluate the decomposition correlation among human annotators by measur-

| Few-shot | | 0.5% | 1% | 5% | 10% |
|---|---|---|---|---|---|
| BART | | 38.29 | 40.77 | 50.30 | 54.23 |
| FT-KGPT* | BLEU | 22.30 | 25.60 | 41.20 | 47.90 |
| CBST* | | 38.74 | **44.40** | **54.98** | **58.78** |
| CG-RL | | **39.31** | 42.74 | 51.05 | 52.96 |
| BART | PA | 33.08 | 31.95 | 38.98 | 40.03 |
| CG-RL | | **37.31** | **34.75** | **40.12** | **41.20** |
| BART | OK | 29.50 | 31.56 | 44.75 | 48.63 |
| CG-RL | | **47.19** | **48.44** | **50.94** | **52.35** |

Table 7: Models performance on few-shot settings. Systems marked with * are from previous work. To ensure a fair comparison with CBST, for this set of experiments CG-RL is trained on BART. Note that the numbers in this table cannot be directly compared to those in the previous tables due to the inclusion of additional domains in the WebNLG 2020 training set.

ing the Normalized Mutual Information (NMI)[14] between the hypothesis and the reference.

Additionally, we require each model to generate predicate decompositions for every input example based on the number of clusters present in the selected reference. We evaluate the generated clusters by computing the NMI metric w.r.t. the reference. The results are shown in Table 6. When the models are trained using CGFULL-k and tested on the in-domain dataset, both of the proposed methods show superior performance compared to CG-Random, with CG-RL outperforming CG-Numerical. However, while not falling too far behind, none of these models achieve the same level of correlation as humans. Similar trends can be observed in the results for the other three testing scenarios, which can be found in Table 17 in Appendix I.

## 5.3 Test on Existing Few-shot Benchmarks

Chen et al. (2020) proposed few-shot splits for WebNLG 2020 by randomly selecting a certain portion of examples from the training set. We compare the performance of our best-performed system CG-RL with pre-trained BARG and two prior works, FT-KGPT (Chen et al., 2020) and CBST (Ke et al., 2022), on these splits.[15] FT-KGPT fine-tunes a Knowledge-grounded Language Model that has been pre-trained on 7M tuples-to-sentence data collected from Wikipedia pages. CBST is a BART-based approach that is tuned using a self-training style, on 0.37M structured data without paired texts collected from GenWiki. The results in Table 7

---

[14]NMI quantifies the mutual information between clusters and is normalized to a value between 0 and 1. A score of 0 indicates no mutual information, while a score of 1 signifies perfect correlation.

[15]Additionally, we present the results of our other proposed approaches in Table 18 in Appendix I.

| | |
|---|---|
| **Input** | \<SUB\> acharya institute of technology \<PRED\> president \<OBJ\> "b.m. reddy";
\<SUB\> acharya institute of technology \<PRED\> city \<OBJ\> bangalore;
\<SUB\> acharya institute of technology \<PRED\> established \<OBJ\> 2000;
\<SUB\> acharya institute of technology \<PRED\> country \<OBJ\> "india";
\<SUB\> acharya institute of technology \<PRED\> campus \<OBJ\> "in soldevanahalli, acharya dr. sarvapalli radhakrishnan road, hessarghatta main road, bangalore – 560090.";
\<SUB\> acharya institute of technology \<PRED\> affiliation \<OBJ\> visvesvaraya technological university |
| **T5** | acharya institute of technology in india has its campus in soldevanahalli, acharya dr. sarvapalli radhakrishnan road, hessarghatta main road, bangalore - 560090. |
| **CG-RL** | acharya institute of technology in india was established in 2000. it is affiliated with visvesvaraya technological university.  acharya institute of technology's campus is located in soldevanahalli, acharya dr. sarvapalli radhakrishnan road, hessarghatta main road, bangalore – 560090. bm reddy is the president of the acharya institute of technology. |
| **Input** | \<SUB\> turkey \<PRED\> leaderName \<OBJ\> ahmet davutoğlu;
\<SUB\> turkey \<PRED\> capital \<OBJ\> ankara;
\<SUB\> turkey \<PRED\> largestCity \<OBJ\> istanbul;
\<SUB\> atatürk monument (izmir) \<PRED\> material \<OBJ\> "bronze";
\<SUB\> turkey \<PRED\> currency \<OBJ\> turkish lira;
\<SUB\> atatürk monument (izmir) \<PRED\> inaugurationDate \<OBJ\> "1932-07-27";
\<SUB\> atatürk monument (izmir) \<PRED\> location \<OBJ\> turkey; |
| **T5** | the atatürk monument, made of bronze, is located in istanbul, turkey, where ahmet davutoglu is the leader. |
| **CG-RL** | the atatürk monument (izmir) is located in turkey where ahmet davutoglu is the leader. the inauguration date of the atatürk monument (izmir), made of bronze, is 1932-07-27. ankara is the capital of turkey where the currency is the turkish lira. istanbul is the largest city in turkey. |
| **Input** | \<SUB\> spain \<PRED\> language \<OBJ\> spanish language;
\<SUB\> ajoblanco \<PRED\> region \<OBJ\> andalusia;
\<SUB\> andalusia \<PRED\> leaderName \<OBJ\> susana díaz;
\<SUB\> ajoblanco \<PRED\> country \<OBJ\> spain;
\<SUB\> spain \<PRED\> ethnicGroup \<OBJ\> spaniards |
| **T5** | ajoblanco is from andalusia where spaniards are an ethnic group. |
| **CG-RL** | ajoblanco originates from the country of spain where spaniards are an ethnic group. susana diaz is the leader of andalusia where ajoblanco is from. spanish is a language spoken in spain. |

Table 8: Cherry-picked examples of input and system-generated texts. Models are trained on CGFULL-2

show that CG-RL outperforms BART across all splits and metrics, except for BLEU on the 10% split. FT-KGPT and CBST only reported BLEU scores. In the 1%, 5%, and 10% splits, CBST outperforms our approach. However, we acknowledge that leveraging task-specific pretraining and self-training-based tuning techniques on our text generator can potentially enhance the few-shot generation performance.

# 6   Limitations and Future Work

We have constructed our benchmark exclusively using WebNLG 2017, as it exhibits several favorable characteristics (see Appendix B). Nonetheless, it would be advantageous to expand the benchmark to include data from a more diverse range of resources. Additionally, we recognize the importance of including multiple languages in the benchmark. The multilingual divisions introduced in WebNLG 2022 were not included in our benchmark due to the presence of training data generated automatically using translation models, which resulted in noisy data. In the future, we aim to expand our benchmark to include high-quality multilingual data resources. Recent work (Axelsson and Skantze, 2023; Yuan and Färber, 2023) explored the zero-shot ability

of LLMs on DTG. Across benchmarks including WebNLG, both studies find that GPT-3 and Chat-GPT achieve lower BLEU scores compared to fine-tuned smaller scale models. In addition, LLMs still face challenges in comprehending the semantic relationships between entities, and the generated text often includes hallucinations. Thus, we did not include LLMs in this study. Due to the limited computational resources, we focused our performance testing on T5-base. However, it is important to evaluate the models in different sizes, including LLMs. We anticipate that tasks with longer inputs/outputs, such as multi-document summarization, may derive even greater benefits from the proposed "solving CG through learning to decompose" idea. In the future, we aim to extend this idea to other tasks.

# Acknowledgments

Lapata gratefully acknowledges the support of the UK Engineering and Physical Sciences Research Council (grant EP/W002876/1). Titov is supported by the Dutch National Science Foundation (NWO Vici VI.C.212.053). We thank Ido Dagan, André Martins, Matthias Lindemann, Antonio Miceli-Barone, Parag Jain, Laura Perez-Beltrachini, Yao Fu, Xue Gong, for inspiring discussions.

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

| | |
|---|---|
| **Input** |  agra airport <PRED> location <OBJ> india;
 agra airport <PRED> runwayLength <OBJ> 1818.0;
 agra airport <PRED> operatingOrganisation <OBJ> indian air force;
 agra airport <PRED> elevationAboveTheSeaLevel_(in_metres) <OBJ> 167.94;
 agra airport <PRED> ICAO_Location_Identifier <OBJ> "viag" |
| **T5** | agra airport is operated by the indian air force and has a runway length of 1818.0 metres. |
| **CG-RL** | agra airport is located in india and its icao location identifier is viag. agra airport, operated by the indian air force, has a runway length of 1818.0. agra airport is elevated 167.94 metres above sea level. |
| **Input** |  attica, indiana <PRED> isPartOf <OBJ> united states;
 attica, indiana <PRED> areaTotal <OBJ> 4.14 (square kilometres);
 attica, indiana <PRED> elevationAboveTheSeaLevel <OBJ> 166.0;
 attica, indiana <PRED> isPartOf_1 <OBJ> logan township, fountain county, indiana;
 attica, indiana <PRED> populationDensity <OBJ> 783.1 (inhabitants per square kilometre); |
| **T5** | attica, indiana, is part of logan township, fountain county, indiana, which has a population of 783.1 inhabitants per square kilometre. |
| **CG-RL** | attica, indiana is part of the united states. attica, indiana, is part of logan township, fountain county, indiana. attica, indiana has a population density of 783.1 inhabitants per square kilometre. attica, indiana is located at 166.0 above sea level. the total area of attica, indiana is 4.14 square kilometres. |
| **Input** |  denmark <PRED> leaderName <OBJ> lars løkke rasmussen;
 european university association <PRED> headquarters <OBJ> brussels;
 school of business and social sciences at the aarhus university <PRED> country <OBJ> denmark;
 denmark <PRED> leaderTitle <OBJ> monarchy of denmark;
 school of business and social sciences at the aarhus university <PRED> affiliation <OBJ> european university association;
 denmark <PRED> religion <OBJ> church of denmark; |
| **T5** | the school of business and social sciences at the aarhus university is affiliated with the european university association which has its headquarters in brussels. |
| **CG-RL** | the school of business and social sciences at the aarhus university is affiliated with the european university association and is located in denmark. the leader of denmark is lars lokke rasmussen. denmark's leader title is the monarchy of denmark. the headquarters of the european university association are in brussels. the religion of denmark is church of denmark. |
| **Input** |  buzz aldrin <PRED> birthPlace <OBJ> glen ridge, new jersey;
 buzz aldrin <PRED> nationality <OBJ> united states;
 buzz aldrin <PRED> was_selected_by_NASA <OBJ> 1963;
 buzz aldrin <PRED> was_a_crew_member_of <OBJ> apollo 11;
 buzz aldrin <PRED> occupation <OBJ> fighter pilot;
 buzz aldrin <PRED> almaMater <OBJ> "massachusetts institute of technology, sc.d. 1963" |
| **T5** | buzz aldrin, who was a member of the apollo 11 crew, was born in glen ridge, new jersey. |
| **CG-RL** | buzz aldrin, who was born in glen ridge, new jersey, was selected by nasa in 1963. buzz aldrin, a us citizen, graduated from massachusetts institute of technology, sc.d. 1963. buzz aldrin, a fighter pilot, was a crew member of apollo 11. |
| **Input** |  turkey <PRED> leaderTitle <OBJ> president of turkey;
 turkey <PRED> leader <OBJ> ahmet davutoğlu;
 atatürk monument (izmir) <PRED> designer <OBJ> pietro canonica;
 turkey <PRED> capital <OBJ> ankara;
 atatürk monument (izmir) <PRED> material <OBJ> "bronze";
 atatürk monument (izmir) <PRED> inaugurationDate <OBJ> "1932-07-27";
 atatürk monument (izmir) <PRED> location <OBJ> turkey; |
| **T5** | the atatürk monument in turkey, made of bronze, was inaugurated on 27th july, 1932. |
| **CG-RL** | the atatürk monument (izmir) is located in turkey where ahmet davutoglu is the leader. the atatürk monument in izmir, made of bronze, was inaugurated on 27th july 1932. ankara is the capital of turkey. the leader of turkey is known as the president of turkey. |
| **Input** |  buzz aldrin <PRED> birthPlace <OBJ> glen ridge, new jersey;
 buzz aldrin <PRED> was_a_crew_member_of <OBJ> apollo 11;
 buzz aldrin <PRED> status <OBJ> "retired";
 buzz aldrin <PRED> nationality <OBJ> united states;
 buzz aldrin <PRED> almaMater <OBJ> "massachusetts institute of technology, sc.d. 1963";
 apollo 11 <PRED> backup_pilot <OBJ> william anders;
 apollo 11 <PRED> operator <OBJ> nasa; |
| **T5** | buzz aldrin, who was a crew member of nasa's apollo 11, was born in glen ridge, new jersey. |
| **CG-RL** | buzz aldrin was born in glen ridge, new jersey. buzz aldrin is from the united states. buzz aldrin, who graduated from massachusetts institute of technology in 1963 with a doctorate in science, is now retired. buzz aldrin was a crew member of apollo 11. apollo 11 was operated by nasa. the backup pilot of apollo 11 was william anders. |

Table 9: Randomly-picked examples of input and system-generated texts. Models are trained on CGFULL-2 and tested with examples equal or more than 5 tuples as input.

## A  Examples of system outputs

## B  Reasons of choosing WebNLG

We choose WebNLG because it offers examples with different input/output sizes. The training set covers multiple domains with diverse predefined predicates. It also includes out-of-domain test examples, allowing us to create challenging scenarios that involve both compositional generalization and cross-domain generalization. Moreover, several faithfulness evaluation metrics have been shown to be highly correlated with human judgments on the WebNLG dataset.

## C  Data Preprocess for Tuple-to-Sentence Alignment

Concretely, we first preprocess the training examples by tokenizing the predicates in the input tuples[16], resolving coreferences in the output texts[17], and splitting the texts into sentences[18].

## D  Transformer-based Weight Predication

The dimension of the model is 128; the dimension of the feedforward is 256; number of heads is 4; number of layers is 2; dropout is 0.1. Other details will be shared with the code.

## E  Human Evaluation Standards

Grammar Correctness
2: The generated text is free of grammar mistakes.
1: The generated text may contain minor grammar mistakes, but they do not significantly affect understanding.
0: The generated text is difficult to understand due to significant grammar mistakes.

Repetition
2: The generated text is natural and free of repetition.
1: The generated text may contain minor instances of repetition, but it is overall natural.
0: The generated text exhibits significant repetition, such as mentioning input triples multiple times or using repetitive sentence structures (e.g., starting sentences with the same entity).

Hallucination
2: The generated text dose not contain any hallucinations.
1: Less than half of the generated text is hallucinated.
0: More than half of the generated text is hallucinated.

Omission
2: The generated text fully covered all the input information.
1: The generated text misses one tuple from the input.
0: The generated text misses more than one tuple from the input.

---

[16]https://pypi.org/project/wordsegment/.
[17]https://github.com/huggingface/neuralcoref.
[18]https://www.nltk.org/api/nltk.tokenize.html.

This section presents an overview of the experimental results. The performance of all models can be found in Table 10 and Table 11. The results for testing scenario 1 and 2 can be found in the left and right sections of Table 10, respectively. Similarly, the results for testing scenario 3 and 4 can be found in the left and right sections of Table 11, respectively.

**CG-NN *vs.* CG-RL**   CG-NN and CG-RL show similar performance overall. CG-RL tends to outperform CG-NN on faithfulness metrics when trained on examples with more input tuples, and on BLEU when trained on examples with fewer tuples. This trend is particularly evident when trained on CGONESHOT (right part of Table 10 and 11).

| SEEN | | CGFULL | | | | CGONESHOT | | | | | |
|---|---|---|---|---|---|---|---|---|---|---|---|
| | | -2 | -3 | -4 | -7 | -2 | -3 | -4 | -5 | -6 | -7 |
| T5 | BLEU | 52.54 | 58.70 | 62.63 | 65.01 | 44.94 | 49.43 | 54.39 | 56.86 | 58.49 | 58.98 |
| CG-Random | | 53.27 | 53.20 | 54.15 | 54.77 | 47.01 | 49.03 | 49.32 | 51.06 | 50.39 | 51.08 |
| CG-Numerical | | 53.40 | 59.21 | 62.03 | 64.77 | 47.46 | 52.91 | 55.37 | **57.81** | 56.32 | 58.07 |
| CG-NN | | 53.87 | **61.36** | **62.75** | **65.04** | 47.28 | 52.79 | 55.73 | 57.60 | **58.68** | **59.14** |
| CG-RL | | **54.56** | 61.02 | 61.78 | 65.01 | **47.58** | **53.17** | **57.06** | 57.41 | 58.08 | 58.44 |
| T5 | PARENT | 54.31 | 60.26 | **61.67** | **62.17** | 46.03 | 50.61 | 53.75 | 54.33 | 55.14 | 56.12 |
| CG-Random | | 56.95 | 59.00 | 60.02 | 60.31 | 50.43 | 53.14 | **55.37** | 55.39 | **56.04** | 56.75 |
| CG-Numerical | | **58.75** | 60.40 | 61.39 | 60.93 | 52.23 | **53.26** | 55.24 | 55.39 | 55.90 | **56.92** |
| CG-NN | | 58.44 | 60.39 | 60.67 | 60.75 | 52.09 | 52.87 | 54.96 | **55.54** | 55.40 | 56.39 |
| CG-RL | | 58.52 | **60.47** | 61.07 | 60.68 | **52.11** | 52.84 | 54.33 | 55.24 | 55.47 | 56.52 |
| T5 | Ok-percent | 43.87 | 68.07 | 76.52 | 78.27 | 36.56 | 47.79 | 64.16 | 68.07 | 74.46 | 78.37 |
| CG-Random | | 69.00 | 78.58 | **81.67** | **82.39** | 59.22 | **68.38** | **77.34** | **79.09** | **81.26** | **81.15** |
| CG-Numerical | | 74.97 | **79.61** | 79.92 | 79.30 | 64.88 | 62.92 | 71.68 | 73.02 | 79.92 | 80.23 |
| CG-NN | | **75.08** | 76.93 | 78.78 | 78.78 | **65.09** | 63.54 | 71.27 | 74.25 | 76.11 | 79.20 |
| CG-RL | | 74.25 | 76.00 | 79.81 | 78.99 | 63.65 | 61.28 | 68.92 | 74.67 | 77.24 | 79.40 |

Table 10: Models performance on *seen* category. The top-performing system is highlighted in bold, while the second best system for Ok-percent is underlined.

| UNSEEN | | CGFULL | | | | CGONESHOT | | | | | |
|---|---|---|---|---|---|---|---|---|---|---|---|
| | | -2 | -3 | -4 | -7 | -2 | -3 | -4 | -5 | -6 | -7 |
| T5 | BLEU | 40.83 | **47.14** | **48.64** | **50.01** | 35.80 | 39.79 | **44.57** | **45.06** | **47.43** | **47.69** |
| CG-Random | | **43.17** | 44.20 | 42.24 | 44.00 | **39.19** | **40.00** | 40.57 | 41.51 | 42.46 | 41.87 |
| CG-Numerical | | 38.78 | 40.46 | 39.39 | 40.52 | 35.31 | 36.83 | 37.66 | 38.21 | 39.83 | 39.13 |
| CG-NN | | 39.78 | 43.99 | 44.81 | 47.69 | 35.65 | 37.42 | 39.81 | 40.99 | 43.23 | 42.95 |
| CG-RL | | 40.42 | 43.61 | 41.81 | 46.21 | 36.06 | 37.94 | 39.86 | 40.73 | 42.38 | 42.84 |
| T5 | PARENT | 42.92 | 49.46 | 49.60 | **51.75** | 37.79 | 40.18 | 44.96 | 46.59 | 47.86 | 48.14 |
| CG-Random | | 46.33 | 49.16 | 49.17 | 49.60 | 41.90 | 43.41 | 46.19 | 48.25 | 48.30 | 48.12 |
| CG-Numerical | | **48.60** | 49.25 | 49.20 | 49.67 | 44.48 | **44.15** | 46.78 | 48.17 | 48.82 | 48.58 |
| CG-NN | | 48.19 | 49.98 | **49.68** | 50.85 | 44.40 | 44.14 | 46.85 | 48.65 | 48.94 | 48.84 |
| CG-RL | | 48.22 | **50.01** | 49.30 | 50.75 | **44.67** | 43.87 | 46.64 | **48.72** | **49.22** | **49.19** |
| T5 | Ok-percent | 43.43 | 56.90 | 68.07 | 67.68 | 34.34 | 41.19 | 53.42 | 55.22 | 57.46 | 64.31 |
| CG-Random | | 64.09 | 67.56 | 73.40 | 76.32 | 52.41 | 55.33 | 63.86 | 64.76 | 66.55 | 70.15 |
| CG-Numerical | | **77.22** | 74.97 | **77.55** | **78.90** | **70.82** | 69.02 | **71.38** | **71.49** | **71.60** | **72.62** |
| CG-NN | | 76.66 | 74.07 | 72.84 | 74.41 | 70.03 | **69.02** | 69.36 | 70.93 | 66.67 | 70.48 |
| CG-RL | | 74.19 | **75.08** | 73.51 | 76.99 | 67.90 | 68.01 | 68.24 | 71.16 | 69.25 | 72.50 |

Table 11: Models performance on *unseen* category.

# G  T5-Large Results

| SEEN | | CGFULL | | | | CGONESHOT | | | | | |
|---|---|---|---|---|---|---|---|---|---|---|---|
| | | -2 | -3 | -4 | -7 | -2 | -3 | -4 | -5 | -6 | -7 |
| T5 | BLEU | 52.54 | 58.70 | 62.63 | 65.01 | 44.94 | 49.43 | 54.39 | 56.86 | 58.49 | 58.98 |
| CG-Random | | 53.27 | 53.20 | 54.15 | 54.77 | 47.01 | 49.03 | 49.32 | 51.06 | 50.39 | 51.08 |
| CG-Numerical | | 53.40 | 59.21 | 62.03 | 64.77 | 47.46 | 52.91 | 55.37 | **57.81** | 56.32 | 58.07 |
| CG-NN | | 53.87 | **61.36** | **62.75** | **65.04** | 47.28 | 52.79 | 55.73 | 57.60 | **58.68** | **59.14** |
| CG-RL | | **54.56** | 61.02 | 61.78 | 65.01 | **47.58** | **53.17** | **57.06** | 57.41 | 58.08 | 58.44 |
| T5 | PARENT | 54.31 | 60.26 | **61.67** | 62.17 | 46.03 | 50.61 | 53.75 | 54.33 | 55.14 | 56.12 |
| CG-Random | | 56.95 | 59.00 | 60.02 | 60.31 | 50.43 | 53.14 | **55.37** | 55.39 | **56.04** | 56.75 |
| CG-Numerical | | **58.75** | 60.40 | 61.39 | 60.93 | 52.23 | **53.26** | 55.24 | 55.39 | 55.90 | **56.92** |
| CG-NN | | 58.44 | 60.39 | 60.67 | 60.75 | 52.09 | 52.87 | 54.96 | **55.54** | 55.40 | 56.39 |
| CG-RL | | 58.52 | **60.47** | 61.07 | 60.68 | **52.11** | 52.84 | 54.33 | 55.24 | 55.47 | 56.52 |
| T5 | Ok-percent | 43.87 | 68.07 | 76.52 | 78.27 | 36.56 | 47.79 | 64.16 | 68.07 | 74.46 | 78.37 |
| CG-Random | | 69.00 | 78.58 | **81.67** | **82.39** | 59.22 | **68.38** | **77.34** | **79.09** | **81.26** | **81.15** |
| CG-Numerical | | 74.97 | **79.61** | 79.92 | 79.30 | 64.88 | 62.92 | 71.68 | 73.02 | 79.92 | 80.23 |
| CG-NN | | **75.08** | 76.93 | 78.78 | 78.78 | **65.09** | 63.54 | 71.27 | 74.25 | 76.11 | 79.20 |
| CG-RL | | 74.25 | 76.00 | 79.81 | 78.99 | 63.65 | 61.28 | 68.92 | 74.67 | 77.24 | 79.40 |

Table 12: T5-Large based models performance on *seen* category. The top-performing system is highlighted in bold, while the second best system for Ok-percent is underlined.

| UNSEEN | | CGFULL | | | | CGONESHOT | | | | | |
|---|---|---|---|---|---|---|---|---|---|---|---|
| | | -2 | -3 | -4 | -7 | -2 | -3 | -4 | -5 | -6 | -7 |
| T5 | BLEU | 40.83 | **47.14** | **48.64** | **50.01** | 35.80 | 39.79 | **44.57** | **45.06** | **47.43** | **47.69** |
| CG-Random | | **43.17** | 44.20 | 42.24 | 44.00 | **39.19** | **40.00** | 40.57 | 41.51 | 42.46 | 41.87 |
| CG-Numerical | | 38.78 | 40.46 | 39.39 | 40.52 | 35.31 | 36.83 | 37.66 | 38.21 | 39.83 | 39.13 |
| CG-NN | | 39.78 | 43.99 | 44.81 | 47.69 | 35.65 | 37.42 | 39.81 | 40.99 | 43.23 | 42.95 |
| CG-RL | | 40.42 | 43.61 | 41.81 | 46.21 | 36.06 | 37.94 | 39.86 | 40.73 | 42.38 | 42.84 |
| T5 | PARENT | 42.92 | 49.46 | 49.60 | **51.75** | 37.79 | 40.18 | 44.96 | 46.59 | 47.86 | 48.14 |
| CG-Random | | 46.33 | 49.16 | 49.17 | 49.60 | 41.90 | 43.41 | 46.19 | 48.25 | 48.30 | 48.12 |
| CG-Numerical | | **48.60** | 49.25 | 49.20 | 49.67 | 44.48 | **44.15** | 46.78 | 48.17 | 48.82 | 48.58 |
| CG-NN | | 48.19 | 49.98 | **49.68** | 50.85 | 44.40 | 44.14 | 46.85 | 48.65 | 48.94 | 48.84 |
| CG-RL | | 48.22 | **50.01** | 49.30 | 50.75 | **44.67** | 43.87 | 46.64 | **48.72** | **49.22** | **49.19** |
| T5 | Ok-percent | 43.43 | 56.90 | 68.07 | 67.68 | 34.34 | 41.19 | 53.42 | 55.22 | 57.46 | 64.31 |
| CG-Random | | 64.09 | 67.56 | 73.40 | 76.32 | 52.41 | 55.33 | 63.86 | 64.76 | 66.55 | 70.15 |
| CG-Numerical | | **77.22** | 74.97 | **77.55** | **78.90** | 70.82 | 69.02 | **71.38** | **71.49** | **71.60** | **72.62** |
| CG-NN | | 76.66 | 74.07 | 72.84 | 74.41 | 70.03 | **69.02** | 69.36 | 70.93 | 66.67 | 70.48 |
| CG-RL | | 74.19 | **75.08** | 73.51 | 76.99 | 67.90 | 68.01 | 68.24 | 71.16 | 69.25 | 72.50 |

Table 13: T5-Large based models performance on *unseen* category.

| SEEN | | CGFULL | | | | CGONESHOT | | | | | |
|---|---|---|---|---|---|---|---|---|---|---|---|
| | | -2 | -3 | -4 | -7 | -2 | -3 | -4 | -5 | -6 | -7 |
| T5 | | 52.54 | 58.70 | 62.63 | 65.01 | 44.94 | 49.43 | 54.39 | 56.86 | 58.49 | 58.98 |
| CG-Random | | 53.27 | 53.20 | 54.15 | 54.77 | 47.01 | 49.03 | 49.32 | 51.06 | 50.39 | 51.08 |
| CG-Numerical | BLEU | 53.40 | 59.21 | 62.03 | 64.77 | 47.46 | 52.91 | 55.37 | **57.81** | 56.32 | 58.07 |
| CG-NN | | 53.87 | **61.36** | **62.75** | **65.04** | 47.28 | 52.79 | 55.73 | 57.60 | **58.68** | **59.14** |
| CG-RL | | **54.56** | 61.02 | 61.78 | 65.01 | **47.58** | **53.17** | **57.06** | 57.41 | 58.08 | 58.44 |
| T5 | | 54.31 | 60.26 | **61.67** | **62.17** | 46.03 | 50.61 | 53.75 | 54.33 | 55.14 | 56.12 |
| CG-Random | | 56.95 | 59.00 | 60.02 | 60.31 | 50.43 | 53.14 | **55.37** | 55.39 | **56.04** | 56.75 |
| CG-Numerical | PARENT | **58.75** | 60.40 | 61.39 | 60.93 | 52.23 | **53.26** | 55.24 | 55.39 | 55.90 | **56.92** |
| CG-NN | | 58.44 | 60.39 | 60.67 | 60.75 | 52.09 | 52.87 | 54.96 | **55.54** | 55.40 | 56.39 |
| CG-RL | | 58.52 | **60.47** | 61.07 | 60.68 | **52.11** | 52.84 | 54.33 | 55.24 | 55.47 | 56.52 |
| T5 | | 43.87 | 68.07 | 76.52 | 78.27 | 36.56 | 47.79 | 64.16 | 68.07 | 74.46 | 78.37 |
| CG-Random | | 69.00 | 78.58 | **81.67** | **82.39** | 59.22 | **68.38** | 77.34 | 79.09 | 81.26 | 81.15 |
| CG-Numerical | OK-percent | 74.97 | **79.61** | 79.92 | 79.30 | 64.88 | 62.92 | 71.68 | 73.02 | 79.92 | 80.23 |
| CG-NN | | **75.08** | 76.93 | 78.78 | 78.78 | **65.09** | 63.54 | 71.27 | 74.25 | 76.11 | 79.20 |
| CG-RL | | 74.25 | 76.00 | 79.81 | 78.99 | 63.65 | 61.28 | 68.92 | 74.67 | 77.24 | 79.40 |

Table 14: T5-Small based models performance on *seen* category. The top-performing system is highlighted in bold, while the second best system for Ok-percent is underlined.

| UNSEEN | | CGFULL | | | | CGONESHOT | | | | | |
|---|---|---|---|---|---|---|---|---|---|---|---|
| | | -2 | -3 | -4 | -7 | -2 | -3 | -4 | -5 | -6 | -7 |
| T5 | | 40.83 | **47.14** | **48.64** | **50.01** | 35.80 | 39.79 | **44.57** | **45.06** | **47.43** | **47.69** |
| CG-Random | | **43.17** | 44.20 | 42.24 | 44.00 | **39.19** | **40.00** | 40.57 | 41.51 | 42.46 | 41.87 |
| CG-Numerical | BLEU | 38.78 | 40.46 | 39.39 | 40.52 | 35.31 | 36.83 | 37.66 | 38.21 | 39.83 | 39.13 |
| CG-NN | | 39.78 | 43.99 | 44.81 | 47.69 | 35.65 | 37.42 | 39.81 | 40.99 | 43.23 | 42.95 |
| CG-RL | | 40.42 | 43.61 | 41.81 | 46.21 | 36.06 | 37.94 | 39.86 | 40.73 | 42.38 | 42.84 |
| T5 | | 42.92 | 49.46 | 49.60 | **51.75** | 37.79 | 40.18 | 44.96 | 46.59 | 47.86 | 48.14 |
| CG-Random | | 46.33 | 49.16 | 49.17 | 49.60 | 41.90 | 43.41 | 46.19 | 48.25 | 48.30 | 48.12 |
| CG-Numerical | PARENT | **48.60** | 49.25 | 49.20 | 49.67 | 44.48 | **44.15** | **46.78** | 48.17 | 48.82 | 48.58 |
| CG-NN | | 48.19 | 49.98 | **49.68** | 50.85 | 44.40 | 44.14 | 46.85 | 48.65 | 48.94 | 48.84 |
| CG-RL | | 48.22 | **50.01** | 49.30 | 50.75 | **44.67** | 43.87 | 46.64 | **48.72** | **49.22** | **49.19** |
| T5 | | 43.43 | 56.90 | 68.07 | 67.68 | 34.34 | 41.19 | 53.42 | 55.22 | 57.46 | 64.31 |
| CG-Random | | 64.09 | 67.56 | 73.40 | 76.32 | 52.41 | 55.33 | 63.86 | 64.76 | 66.55 | 70.15 |
| CG-Numerical | OK-percent | **77.22** | 74.97 | **77.55** | **78.90** | **70.82** | **69.02** | **71.38** | **71.49** | **71.60** | **72.62** |
| CG-NN | | 76.66 | 74.07 | 72.84 | 74.41 | 70.03 | **69.02** | 69.36 | 70.93 | 66.67 | 70.48 |
| CG-RL | | 74.19 | **75.08** | 73.51 | 76.99 | 67.90 | 68.01 | 68.24 | 71.16 | 69.25 | 72.50 |

Table 15: T5-Small based models performance on *unseen* category.

# I Extra Experiment Results

| Avg #Clusters | | CGFull | | | | CGOneShot | | | | | |
|---|---|---|---|---|---|---|---|---|---|---|---|
| | | -2 | -3 | -4 | -7 | -2 | -3 | -4 | -5 | -6 | -7 |
| T5 | SEEN | 1.07 | 1.19 | 1.46 | 1.38 | 1.05 | 1.09 | 1.26 | 1.4 | 1.44 | 1.42 |
| CG-Random | | 2.04 | 2.03 | 2.00 | 2.04 | 1.98 | 1.98 | 2.02 | 2.03 | 2.00 | 2.00 |
| CG-Numerical | | 2.05 | 1.59 | 1.25 | 1.13 | 2.01 | 1.53 | 1.34 | 1.27 | 1.23 | 1.22 |
| CG-NN | | 2.00 | 1.36 | 1.11 | 1.06 | 2.01 | 1.53 | 1.3 | 1.28 | 1.09 | 1.08 |
| CG-RL | | 1.91 | 1.38 | 1.23 | 1.07 | 1.99 | 1.47 | 1.22 | 1.28 | 1.16 | 1.13 |
| T5 | UNSEEN | 1.02 | 1.15 | 1.40 | 1.32 | 1.01 | 1.02 | 1.26 | 1.29 | 1.37 | 1.4 |
| CG-Random | | 1.88 | 1.85 | 1.87 | 1.88 | 1.90 | 1.89 | 1.87 | 1.82 | 1.87 | 1.82 |
| CG-Numerical | | 2.61 | 2.46 | 2.43 | 2.41 | 2.62 | 2.47 | 2.43 | 2.43 | 2.42 | 2.42 |
| CG-NN | | 2.44 | 2.00 | 1.56 | 1.45 | 2.58 | 2.37 | 2.00 | 2.05 | 1.79 | 1.74 |
| CG-RL | | 2.34 | 2.04 | 1.91 | 1.63 | 2.50 | 2.24 | 1.96 | 2.09 | 1.93 | 1.84 |

Table 16: The average number of groups each model generates for test examples. The average number of sentences for human written references are 1.37 and 1.35 for seen and unseen test set respectively.

| NMI | | CGFull | | | | CGOneShot | | | | | |
|---|---|---|---|---|---|---|---|---|---|---|---|
| | | -2 | -3 | -4 | -7 | -2 | -3 | -4 | -5 | -6 | -7 |
| CG-Random | SEEN | 40.87 | 38.59 | 43.55 | 36.02 | 39.43 | 38.43 | 40.74 | 39.06 | 37.20 | 35.76 |
| CG-Numerical | | 44.87 | 52.35 | 57.03 | 60.64 | 46.72 | 55.71 | 59.79 | 62.60 | 61.81 | 60.65 |
| CG-NN | | 47.08 | 58.87 | 65.10 | 66.24 | 48.03 | 58.97 | 65.49 | 64.40 | 66.18 | 66.18 |
| CG-RL | | 47.55 | 62.17 | 62.51 | 64.33 | 48.08 | 56.82 | 65.21 | 63.17 | 64.56 | 63.39 |
| CG-Random | UNSEEN | 44.18 | 39.43 | 40.59 | 37.01 | 41.20 | 39.71 | 37.21 | 39.11 | 38.55 | 38.34 |
| CG-Numerical | | 41.78 | 39.04 | 41.17 | 41.83 | 38.20 | 45.10 | 41.19 | 47.76 | 42.57 | 41.63 |
| CG-NN | | 50.95 | 53.17 | 55.30 | 52.60 | 52.63 | 52.26 | 52.40 | 54.59 | 55.57 | 52.03 |
| CG-RL | | 47.77 | 53.19 | 49.66 | 47.20 | 57.28 | 52.73 | 53.79 | 53.57 | 54.01 | 49.65 |

Table 17: Predicate decomposition performance of all models evaluated using NMI. The NMI among human annotated references are 70.06 and 67.31 for seen and unseen test set respectively. Since we can not control the amount of sentences the vanilla T5 generates, we discard the T5 for this experiment.

| Few-shot | | 0.5% | 1% | 5% | 10% |
|---|---|---|---|---|---|
| BART | | 38.29 | 40.77 | 50.30 | 54.23 |
| FT-KGPT* | | 22.30 | 25.60 | 41.20 | 47.90 |
| CBST* | **BLEU** | 38.74 | **44.40** | **54.98** | **58.78** |
| CG-Random | | **39.93** | 42.50 | 48.31 | 49.50 |
| CG-Numerical | | 39.43 | 43.37 | 51.12 | 53.96 |
| CG-NN | | 39.63 | 43.96 | 50.9 | 53.95 |
| CG-RL | | 39.31 | 42.74 | 51.05 | 52.96 |
| BART | | 33.08 | 31.95 | 38.98 | 40.03 |
| CG-Random | | 36.39 | 33.76 | 38.80 | 39.69 |
| CG-Numerical | **PARENT** | **37.50** | 34.68 | **40.22** | 40.84 |
| CG-NN | | 37.48 | 34.59 | 39.87 | 40.93 |
| CG-RL | | 37.31 | **34.75** | 40.12 | **41.20** |
| BART | | 29.50 | 31.56 | 44.75 | 48.63 |
| CG-Random | | 41.44 | 43.56 | **52.50** | 51.94 |
| CG-Numerical | **OK-percent** | **47.63** | 45.31 | 50.00 | 51.50 |
| CG-NN | | 47.31 | 44.37 | 50.06 | 51.25 |
| CG-RL | | 47.19 | **48.44** | 50.94 | **52.35** |

Table 18: Models performance on few-shot settings. Systems marked with * are from previous work. To ensure a fair comparison with CBST, for this set of experiments all CG- models are trained on BART. Note that the numbers in this table cannot be directly compared to those in the previous tables due to the inclusion of additional domains in the WebNLG 2020 training set.