# OpenReview forum: "Compositional Generalization for Data-to-Text Generation"
_EMNLP/2023/Conference — EMNLP 2023 Findings_

### Official Review · Reviewer_7idM · 2023-08-04

**Typos Grammar Style And Presentation Improvements:** N/A
**Soundness:** 4

**Excitement:**

3: Ambivalent: It has merits (e.g., it reports state-of-the-art results, the idea is nice), but there are key weaknesses (e.g., it describes incremental work), and it can significantly benefit from another round of revision. However, I won't object to accepting it if my co-reviewers champion it.

**Missing References:**

N/A

**Paper Topic And Main Contributions:**

This paper focuses on assessing the compositional generalization ability on the Data-to-Text generation task. The authors propose a benchmark based on WebNLG 2017 to specifically assess the compositionality in four different scenarios. The authors present an innovative framework (predicate decomposition) which demonstrates improved performance compared to T5. Beyond the standard BLEU evaluation, this paper brings more attention to the faithfulness evaluation (PARENT and OK-percent).

**Questions For The Authors:**

Question A: There are sorted and grouped predicates in the training set of WebNLG. Have you considered using them in training?

**Reasons To Accept:**

It is intuitive to decompose predicates into groups first (trying to ensure at least one combination pair has been seen) and then generate text for each group. The results show improvements over T5 under different settings

The benchmark (especially the few-shot one) can be useful in this field.

Ablation study and analysis experiments (along discussions) are clear and in detail.

**Reasons To Reject:**

The CG-Random is performing quite well in many scenarios. This brings doubts on the effect of the complex predicate composition process.


**Reproducibility:**

4: Could mostly reproduce the results, but there may be some variation because of sample variance or minor variations in their interpretation of the protocol or method.

**Reviewer Confidence:**

2: Willing to defend my evaluation, but it is fairly likely that I missed some details, didn't understand some central points, or can't be sure about the novelty of the work.

---

> ### Author Rebuttal · Authors · 2023-08-29
>
> We thank the reviewer's efforts.
>
> ## RE-1: CG-Random is enough
>
> It is important to highlight that our paper introduces **four** distinct testing scenarios. Our objective is to identify methods that excel across **all of them**. We assume a method is performant if it can generate text which resembles human writing (as evaluated by BLEU) **and** is also faithful to the input (as evaluated by PARENT and OK-percent).
>
> We next emphasize why CG-Random underperforms according to these criteria:
>
> * In the context of domain adaptation (scenarios 3 and 4), CG-Random displays worse yet somewhat comparable performance to our proposed methods based on BLEU and PARENT metrics, while it is substantially inferior (by **more than 10 points**) according to OK-percent metrics. Please see the lower sections in Table 3 and Table 11). This indicates that our methods produce considerably fewer omissions and hallucinations, while maintaining human-like fluency.
>
> * Regarding in-domain training (scenarios 1 and 2), CG-Random performs slightly worse according to PARENT metrics. This performance gap is much wider when looking at BLEU scores (our methods outperform CG-Random with a margin of **nearly 10 points**). Please see the upper section of Table 2 and Table 10. The sole metric on which CG-Random surpasses our methods is OK-percent, although the difference is marginal. In Section 4.2 (Lines 470-490), we discuss why this is the case and highlight that CG-Random's excessive decompositions might result in faithful generation, but the  approach also leads to unnatural and redundant textual descriptions.
>
> In summary, across **all four testing scenarios**, our proposed methods strike a **better balance** between human-likeness and faithfulness compared to CG-Random without introducing significantly greater complexity. The fundamental generation model remains consistent with T5. When compared to CG-Random, the CG-Numerical method simply requires tallying of the co-occurrences of two predicates within the training set. Similarly, CG-NN only mandates a compact classification network, comprising less than 20 lines of code in its forward pass. As for CG-RL, it can be understood as the integration of the original, unaltered T5 generator with the newly introduced classifier.
>
> In the final version of the paper, we will discuss the limitations of the CG-Random approach more thoroughly.
>
>
> ## RE-2: Use the annotated grouping information in training
>
> Predicates are indeed grouped in the WebNLG 2017 v1.6 dataset. In section 5.2, we utilize this annotation in the test set to assess the decomposition. We assume that such high quality annotations are not usually available for training, and thus design our models around this premise.

---

### Official Review · Reviewer_Yzid · 2023-08-04

**Soundness:** 4

**Excitement:**

4: Strong: This paper deepens the understanding of some phenomenon or lowers the barriers to an existing research direction.

**Paper Topic And Main Contributions:**

This paper presents work in the area of data-to-text generation, which involves the generation of descriptions from tuples (comprising arguments and predicates) that should contain all Information required for accurate text generation. The problem space is the issue of unseen predicate combinations, which requires a model to generalize. Specifically, the authors aim to address compositional generalization, where data-to-text systems produce unfaithful descriptions when faced with unseen predicate combinations.

Their work makes two contributions: ((1) an evaluation method for assessing how existing data-to-text generation algorithms handle compositional generalization; and (2) a novel approach to handling compositional generalization by clustering predicates into groups. Their evaluation method involves training with fewer input tuples than are used during testing; reducing to a single input example for each predicate combination; as well as testing on unseen domains. Under their test environment, they find that SoTA pre-trained language models handle compositional generalization poorly.

They then present an approach to address compositional generalization by clustering predicates into groups. The groups are composed of decomposed unfamiliar predicate combinations that are clustered based on graph weights learned during training. Text is then generated, sentence by sentence, individually for each group of predicates at a time, and then combined to form the final output.


**Reasons To Accept:**

They present a well written and clear paper that shows clearly how their cluster-based compositional generalization method was developed. The intuition to Decompose unseen predicate combinations into smaller groups, ensure that a combination in each group has been seen during training, then generate a sentence from each group individually first, followed by an aggregation, is quite brilliant!

Additionally, it is always advantageous to have improved performance not dependent on more training examples, and their cluster-based generation method requires small training sets, with no need for further labeled or unlabeled data.

Finally, their approach performs very well when evaluated on T5 baselines, and the method of evaluation as well as the discussion of results is clear and relevant.


**Reasons To Reject:**

I find no reason to reject this paper.


**Reproducibility:**

4: Could mostly reproduce the results, but there may be some variation because of sample variance or minor variations in their interpretation of the protocol or method.

**Reviewer Confidence:**

4: Quite sure. I tried to check the important points carefully. It's unlikely, though conceivable, that I missed something that should affect my ratings.

**Typos Grammar Style And Presentation Improvements:**

Caption for Algorithm 1, the spelling of "decomposition"

---

> ### Author Rebuttal · Authors · 2023-08-29
>
> We appreciate the reviewer's positive feedback. We agree with the reviewer that a notable advantage of our approaches lies in its avoidance of the need to acquire extensive datasets. Instead, it centers on the efficient utilization of available resources, which are often modest in scale.

---

### Official Review · Reviewer_GZ7d · 2023-08-08

**Soundness:** 3

**Excitement:**

3: Ambivalent: It has merits (e.g., it reports state-of-the-art results, the idea is nice), but there are key weaknesses (e.g., it describes incremental work), and it can significantly benefit from another round of revision. However, I won't object to accepting it if my co-reviewers champion it.

**Paper Topic And Main Contributions:**

A benchmark to assess the compositional generalization of data-to-text systems is proposed. Authors also present a technique that decomposes unseen structured data combinations into smaller, seen ones. The decomposition idea also used to develop evaluation criteria.

**Reasons To Accept:**

Compositional generalization is an important area of data-to-text generation, having a targeted benchmark to assess this capability is great. It allows testing compositional generalization, few-shot learning, domain adaptation and their combinations.


**Reasons To Reject:**

As already mentioned in the limitations section, the paper uses only T5-base, which is a fairly small model.
For very large models, WebNLG is a fairly easy task, and they will probably already be excellent at compositional generalization. Given LLMs are now SoTA and ubiquitous, the paper seems incomplete without evaluating them.

In the presence of LLMs and distillation based approaches, its not clear if the decomposition based approach offers additional value. For example, you could create all/many combinations of input predicates, use an LLM to get outputs and then train a smaller model on this silver data. Compositional generalization would be much less of an issue, since the small model has been trained on a much larger set of combinations.

**Reproducibility:**

3: Could reproduce the results with some difficulty. The settings of parameters are underspecified or subjectively determined; the training/evaluation data are not widely available.

**Reviewer Confidence:**

2: Willing to defend my evaluation, but it is fairly likely that I missed some details, didn't understand some central points, or can't be sure about the novelty of the work.

---

> ### Author Rebuttal · Authors · 2023-08-29
>
> We thank the reviewers comments.
>
> ## RE-1: Limited to a single model size, T5-base
> Please refer to our response **RE-1: Limited to a single model size** to Reviewer-1.
>
> ## RE-2: Incomplete work without LLMs
> First, it is essential to emphasize that conducting experiments on models of smaller scales still remains valuable. While LLMs are powerful models, they are not universally suitable for all application scenarios, such as [mobile devices](https://www.computerweekly.com/news/366545122/Qualcomm-Meta-collaborate-to-enable-on-device-AI-applications-using-Llama-2) with constrained GPU resources.
>
> Meanwhile, we are not aware of any studies corroborating claims that Data-to-Text tasks like WebNLG are easy for LLMs, and that compositional generalization poses no challenges for them. Recent work (published in July and August, yet to undergo peer review) explores the zero-shot capabilities of LLMs on Data-to-Text tasks: https://arxiv.org/pdf/2307.07312.pdf, https://arxiv.org/pdf/2307.14712.pdf. Across benchmarks including WebNLG, both studies find that GPT-3 and ChatGPT achieve lower BLEU scores compared to fine-tuned smaller scale models. In addition, large-scale LLMs still face challenges in comprehending the semantic relationships between entities. Furthermore, the generated text often includes instances of hallucinations or irrelevant information.
>
> We will discuss both studies in the future work section.
>
> ## RE-3: Distillation based approaches
>
> Data Augmentation serves as one approach to tackle compositional generalization problems, but it is not the sole solution. Other methodologies, such as input decomposition, also warrant investigation.
>
> Knowledge distillation, such as Data Augmentation, aims to enhance a model's proficiency in text generation by enlarging the constrained pool of human-annotated training data. Conversely, our approach takes an alternative perspective, focusing on the efficient utilization of the limited yet high-quality annotated resources available.
>
> In contrast to knowledge distillation, our methods necessitate minor adjustments in data preparation, model architecture, and training mechanisms. However, in comparison to our approaches, knowledge distillation also exhibits limitations:
>
> * Generating possible predicate permutations is a laborious process. Within the WebNLG 2017 dataset, each domain typically encompasses an average of 20 distinct predicates. With a maximum of 7 tuples per input, the potential permutations of predicates within each domain, accounting for predicate order, can be calculated as 20 + 20^2 + 20^3 + 20^4 + 20^5 + 20^6 + 20^7. It's important to note that both generating augmented data and fine-tuning a smaller model on this extensive dataset consumes a substantial amount of time.
>
> * Data Augmentation assumes that LLMs possess the capability to produce silver-standard data that resembles human writing and remains faithful to the input. However, as previously noted, recent investigations have not provided substantial evidence to support this particular capability.
>
> * In contrast to the data augmentation strategy, our suggested input decomposition approach is more intuitive and closely aligned with human behaviour. In tasks like Data-to-Text generation, humans aren't required to encounter every conceivable combination of predicates and entities. Through simple demonstrations, when presented with a set of tuples, humans can efficiently break them down into groups and cohesively describe them within multiple sentences.
>
> We plan to incorporate a discussion that addresses the distinctions between our approach and knowledge distillation.

---

### Official Review · Reviewer_1UpM · 2023-08-12

**Soundness:** 4

**Excitement:**

3: Ambivalent: It has merits (e.g., it reports state-of-the-art results, the idea is nice), but there are key weaknesses (e.g., it describes incremental work), and it can significantly benefit from another round of revision. However, I won't object to accepting it if my co-reviewers champion it.

**Paper Topic And Main Contributions:**

The paper delves into the compositional generalization challenges faced by current Data-to-Text models, especially concerning predicate structures. The evaluations revolve around predicate structure-controlled instances, sourced from the WebNLG benchmark. The authors present a methodology that begins with sentence planning -- clustering input predicates where each cluster maps to a subsequent sentence -- followed by the text generation. Various variants of this method are implemented, tested, and contrasted with relevant baselines. The experimental findings indicate that breaking down the predicate structure into smaller chunks heightens the faithfulness of the output and bolsters the model's domain adaptability.

**Questions For The Authors:**

Question A: Given the distinct sentence generation process in CG methods, repetition becomes an evident concern. When entities recur across sentences, the use of pronouns is minimized, leading to redundancy. This issue might escalate when crafting lengthier content, like full-length documents. How do the authors propose to tackle this repetition challenge?

**Reasons To Accept:**

* The evaluations are carefully structured, grounded in predicate configurations. This approach establishes a solid benchmark that future investigations can reference and build upon.
* The paper introduces multiple predictors for predicate decomposition: numerical-based, neural network-driven, and those enhanced by REINFORCE. A random predictor also serves as a baseline in the comparison.
* The paper provides a comprehensive analysis of the experiment results, showing intricacies and broader implications of the CG methods.

**Reasons To Reject:**

* The study is limited to a single model size, leaving open questions about the method's efficacy in bridging compositional generalization gaps in larger models.
* While the CG method seems to exhibit potential in minimizing hallucinations or omissions for longer text, it appears prone to repetition issues, even in WebNLG content that spans an average of 1 to 2 sentences. Effective solutions to this concern aren't elaborated upon.
* The human evaluation (as seen in Table 5) implies that the CG method notably diminishes omissions, given the low scores associated with T5 outputs. This improvement would be more evident if reflected in the OK-percent metric. An inclusion of the 4-way OK-percent metric results would be a valuable addition.


**Reproducibility:**

5: Could easily reproduce the results.

**Reviewer Confidence:**

4: Quite sure. I tried to check the important points carefully. It's unlikely, though conceivable, that I missed something that should affect my ratings.

---

> ### Author Rebuttal · Authors · 2023-08-29
>
> We thank the reviewer's suggestions.
>
> ## RE-1: Limited to a single model size
>
> We will incorporate an additional subsection within Section 5 (In-depth Analysis and Discussion), dedicated to investigating how our predicate decomposition approach  interacts with model size (T5-small, T5-base, T5-Large). We are including partial outcomes from **T5-Large** in this reply. Due to the time constraints of the rebuttal stage, we were unable to carry out all experiments. The concluded outcomes **exhibit a very similar trend** to those of the T5-base model. Complete experimental findings for both T5-Small and T5-Large will be available in the final edition of this paper.  *(Note: OK-percent Omi stands for the amount of omissions observed in the system generations. The lower OK-percent Omi the better)*
>
> * Performance of T5-Large-based models evaluated in scenario 2 (refer to Figure 3, corresponding to the Table 2 in the paper), i.e. trained on CGOneShot-k and tested on SEEN category.
> |CGOneShot-Seen|Models| -2 | -3 | -4 |
> | -------- | -------- | ------- | ------- | -------- |
> |BLEU|T5-Large | 46.55 | 50.12 | 55.82 |
> ||CG-Random | 46.89 | 48.27 | 50.15 |
> ||CG-Numerical | 47.73 | 53.11 | 56.09 |
> ||CG-NN | 47.59 | 52.73 | 56.73 |
> ||CG-RL | running | running | running |
> |PARENT|T5-Large | 47.95 | 49.94 | 54.44 |
> ||CG-Random | 51.44 | 51.95 | 55.16 |
> ||CG-Numerical | 53.32 | 52.97 | 55.53 |
> ||CG-NN | 53.25 | 52.72 | 55.44 |
> ||CG-RL | running | running | running |
> |OK-percent|T5-Large | 38.41 | 50.36 | 66.43 |
> ||CG-Random | 59.73 | 67.66 | 78.17 |
> ||CG-Numerical | 67.97 | 65.60 | 74.87 |
> ||CG-NN | 67.04 | 65.71 | 73.43 |
> ||CG-RL | running | running | running |
> |OK-percent (Omi)|T5-Large | 53.14 | 42.53 | 29.25 |
> ||CG-Random | 33.57 | 25.33 | 16.89 |
> ||CG-Numerical | 26.16 | 27.81 | 27.81 |
> ||CG-NN | 27.19 | 27.60 | 22.66 |
> ||CG-RL | running | running | running |
>
> * Performance of T5-Large-based models evaluated in scenario 4 (refer to Figure 3, corresponding to the Table 3 in the paper), i.e. trained on CGOneShot-k and tested on UNSEEN category.
> |CGOneShot-Unseen|Models| -2 | -3 | -4 |
> | -------- | -------- | ------- | ------- | -------- |
> |BLEU|T5-Large | 38.05 | 41.44 | 46.75 |
> ||CG-Random | 41.13 | 41.96 | 42.55 |
> ||CG-Numerical | 36.88 | 38.66 | 39.61 |
> ||CG-NN | 36.97 | 38.82 | 41.96 |
> ||CG-RL | running | running | running |
> |PARENT|T5-Large | 41.18 | 43.12| 46.72 |
> ||CG-Random | 44.26 | 45.26 | 46.54 |
> ||CG-Numerical | 45.77 | 45.73 | 47.07 |
> ||CG-NN | 45.76 | 45.61 | 47.54 |
> ||CG-RL | running | running | running |
> |OK-percent|T5-Large | 36.48 | 43.32 | 56.45 |
> ||CG-Random | 52.97 | 57.80 | 66.22 |
> ||CG-Numerical | 66.78 | 63.97 | 70.59 |
> ||CG-NN | 66.33 | 63.52 | 70.15 |
> ||CG-RL | running | running | running |
> |OK-percent (Omi)|T5-Large | 46.91 | 41.75 | 31.99 |
> ||CG-Random | 32.44 | 27.83 | 22.78 |
> ||CG-Numerical | 19.87| 24.24 | 19.30 |
> ||CG-NN | 19.87 | 23.57 | 19.64 |
> ||CG-RL | running | running | running |
>
> * Performance of T5-Large-based models evaluated in scenario 1 (refer to Figure 3, corresponding to the CGFull section in Table 10 in the paper), i.e. trained on CGFull-k and tested on SEEN category.
> |CGFull-Seen|Models| -2 | -3 | -4 |
> | -------- | -------- | ------- | ------- | -------- |
> |BLEU|T5-Large | 46.55 | 50.12 | 55.82 |
> ||CG-Random | 46.89 | 48.27 | 50.15 |
> ||CG-Numerical | 47.73 | 53.11 | 56.09 |
> ||CG-NN | 47.59 | 52.73 | 56.73 |
> ||CG-RL | running | running | running |
> |PARENT|T5-Large | 47.95 | 49.94| 54.44 |
> ||CG-Random | 51.44 | 51.95 | 55.16 |
> ||CG-Numerical | 53.32 | 52.97 | 55.53 |
> ||CG-NN | 53.25 | 52.72 | 55.44 |
> ||CG-RL | running | running | running |
> |OK-percent|T5-Large | 38.41 | 50.36 | 66.43 |
> ||CG-Random | 59.73 | 67.66 | 78.17 |
> ||CG-Numerical | 67.97 | 65.6 | 74.87 |
> ||CG-NN | 67.04 | 65.71 | 73.43 |
> ||CG-RL | running | running | running |
> |OK-percent (Omi)|T5-Large | 53.14 | 42.53 | 29.25 |
> ||CG-Random | 33.57 | 25.33 | 16.89 |
> ||CG-Numerical | 26.16 | 27.81 | 21.63 |
> ||CG-NN | 27.19 | 27.6 | 22.66 |
> ||CG-RL | running | running | running |
>
> * Performance of T5-Large-based models evaluated in scenario 3 (refer to Figure 3, corresponding to the CGFull section in Table 11 in the paper), i.e. trained on CGFull-k and tested on UNSEEN category.
> |CGFull-Unseen|Models| -2 | -3 | -4 |
> | -------- | -------- | ------- | ------- | -------- |
> |BLEU|T5-Large | 38.05 | 41.44 | 46.75 |
> ||CG-Random | 41.13 | 41.96 | 42.55 |
> ||CG-Numerical | 36.88 | 38.66 | 39.61 |
> ||CG-NN | 36.97 | 38.82 | 41.96 |
> ||CG-RL | running | running | running |
> |PARENT|T5-Large | 41.18 | 43.12 | 46.72 |
> ||CG-Random | 44.26 | 45.26 | 46.54 |
> ||CG-Numerical | 45.77 | 45.73 | 47.07 |
> ||CG-NN | 45.76 | 45.61 | 47.54 |
> ||CG-RL | running | running | running |
> |OK-percent|T5-Large | 36.48 | 43.32 | 56.45 |
> ||CG-Random | 52.97 | 57.8 | 66.22  |
> ||CG-Numerical | 66.78 | 63.97 | 70.59 |
> ||CG-NN | 66.33 | 63.52 | 70.15 |
> ||CG-RL | running | running | running |
> |OK-percent (Omi)|T5-Large | 46.91 | 41.75 | 31.99 |
> ||CG-Random | 32.44 | 27.83 | 22.78 |
> ||CG-Numerical | 19.87 | 24.24 | 19.3 |
> ||CG-NN | 19.87 | 23.57 | 19.64 |
> ||CG-RL | running | running | running |
>
>
> ## RE-2: 4-way OK-percent metric results
> Indeed, the proposed methods receive low omission scores across all testing scenarios when employing the OK-percent metric. We will include the full 4-way OK-percent results in the final version of the paper. We only report the omission scores here in the response.
>
> * Evaluation scenario 2
> |CGOneShot-Seen|Models| -2 | -3 | -4 | -5 | -6 | -7 |
> | -------- | -------- | ------- | ------- | -------- | -------- | -------- | -------- |
> |OK-percent (Omi)|T5-base | 50.77 | 44.59 | 32.03 | 28.63 | 22.45 | 17.51 |
> ||CG-Random | 45.62 | 35.32 | 25.75 | 21.22 | 18.64 | 15.96 |
> ||CG-Numerical | 44.9 | 33.88 | 23.48 | 20.8 | 16.27 | 15.04 |
> ||CG-NN | 45.42 | 33.68 | 23.58 | 19.67 | 16.58 | 15.55 |
> ||CG-RL | 46.14 | 33.57 | 23.17 | 19.05 | 16.89 | 15.24 |
>
> * Evaluation scenario 4
> |CGOneShot-Unseen|Models| -2 | -3 | -4 | -5 | -6 | -7 |
> | -------- | -------- | ------- | ------- | -------- | -------- | -------- | -------- |
> |OK-percent (Omi)|T5-base | 50.28 | 48.04 | 37.82 | 37.49 | 33.33 | 28.28 |
> ||CG-Random | 36.36 | 31.65 | 24.8 | 24.47 | 25.7 | 21.77 |
> ||CG-Numerical | 20.76 | 20.88 | 18.97 | 21.1 | 19.53 | 19.3 |
> ||CG-NN | 21.44 | 20.31 | 20.76 | 22.56 | 25.59 | 22.67 |
> ||CG-RL | 44.11 | 39.06 | 28.84 | 30.75 | 27.5 | 24.58 |
>
> * Evaluation scenario 1
> |CGFull-Seen|Models| -2 | -3 | -4 | -7 |
> | -------- | -------- | ------- | ------- | -------- | -------- |
> |OK-percent (Omi)|T5-base | 50.15 | 27.7 | 18.13 | 16.07 |
> ||CG-Random | 26.78 | 50.77 | 11.12 | 12.26 |
> ||CG-Numerical | 39.96 | 16.27 | 14.42 | 15.04 |
> ||CG-NN | 18.95 | 17.92 | 16.17 | 15.55 |
> ||CG-RL | 20.19 | 19.57 | 14.73 | 15.55 |
>
> * Evaluation scenario 3
> |CGFull-Unseen|Models| -2 | -3 | -4 | -7 |
> | -------- | -------- | ------- | ------- | -------- | -------- |
> |OK-percent (Omi)|T5-base | 52.15 | 36.25 | 16.07 | 27.38|
> ||CG-Random | 40.07 | 26.6 | 22.11 | 21.44 |
> ||CG-Numerical | 39.96 | 24.92 | 22.22 | 22.67 |
> ||CG-NN | 36.03 | 25.36 | 21.21 | 21.89 |
> ||CG-RL | 37.26 | 26.26 | 21.55 | 21.89 |
>
>
> ## RE-3: The repetition challenge
> At present, sentence generation for each predicate group operates autonomously. Consequently, the generation of a new sentence occurs without any awareness of the previously generated sentence, leading to repeated entities. In principle, we can **reintroduce conditional dependencies** between generated sentences. Nonetheless, this approach necessitates careful and thorough experimentation.
> In addition, there exists rapid solutions. For instance, a rule-based system could be developed to identify recurring entities and then assign a likelihood for substitution with references.

---

### Meta-Review · Area_Chair_egAn · 2023-09-18

**Recommendation:** 3

**Metareview:**

The paper investigates the challenges of compositional generalization in Data-to-Text models, particularly in relation to predicate structures. The authors propose a new methodology that begins with sentence planning, whereby input predicates are clustered (each cluster mapping to a subsequent sentence), followed by text generation. The approach is evaluated using predicate structure-controlled instances from the WebNLG benchmark. The experimental results indicate that decomposing the predicate structure into smaller units improves the faithfulness of the output and enhances the model's domain adaptability.

Main Contributions:

The authors present a novel evaluation methodology centered on predicate configurations, providing a robust benchmark for future research.
The paper introduces multiple predictors for predicate decomposition, including numerical-based, neural network-driven, and those enhanced by REINFORCE. A random predictor also serves as a baseline.
The authors propose an innovative framework (predicate decomposition) which demonstrates improved performance compared to T5.
The paper provides a comprehensive analysis of the experiment results, highlighting the nuances and broader implications of the compositional generalization methods.
The authors create a benchmark for assessing compositional generalization, few-shot learning, and domain adaptation, which can be useful in this field.
Reasons for Acceptance:

The paper is well-written and clearly explains the development and implications of their cluster-based compositional generalization method.
The proposed method requires smaller training sets and does not require additional labeled or unlabeled data.
Their approach performs well when evaluated against T5 baselines, and the method of evaluation and discussion of results is clear and relevant.
The introduction of an evaluation method and a novel approach to address compositional generalization fills a gap in the current research.
Reasons for Rejection:

The study is limited to a single model size, leaving questions about the method's efficacy in bridging compositional generalization gaps in larger models.
The proposed method seems prone to repetition issues, particularly in WebNLG content that spans an average of 1 to 2 sentences, and effective solutions to this concern aren't elaborated upon.
The CG-Random performs quite well in many scenarios, raising questions about the necessity and effectiveness of the complex predicate composition process.

---

### Decision · Program_Chairs · 2023-10-07

**Decision:**

Accept-Findings

**Comment:**

The paper investigates the challenges of compositional generalization in Data-to-Text models, particularly in relation to predicate structures. The authors propose a new methodology that begins with sentence planning, whereby input predicates are clustered (each cluster mapping to a subsequent sentence), followed by text generation. The approach is evaluated using predicate structure-controlled instances from the WebNLG benchmark. The experimental results indicate that decomposing the predicate structure into smaller units improves the faithfulness of the output and enhances the model's domain adaptability.

Main Contributions:

The authors present a novel evaluation methodology centered on predicate configurations, providing a robust benchmark for future research.
The paper introduces multiple predictors for predicate decomposition, including numerical-based, neural network-driven, and those enhanced by REINFORCE. A random predictor also serves as a baseline.
The authors propose an innovative framework (predicate decomposition) which demonstrates improved performance compared to T5.
The paper provides a comprehensive analysis of the experiment results, highlighting the nuances and broader implications of the compositional generalization methods.
The authors create a benchmark for assessing compositional generalization, few-shot learning, and domain adaptation, which can be useful in this field.
Reasons for Acceptance:

The paper is well-written and clearly explains the development and implications of their cluster-based compositional generalization method.
The proposed method requires smaller training sets and does not require additional labeled or unlabeled data.
Their approach performs well when evaluated against T5 baselines, and the method of evaluation and discussion of results is clear and relevant.
The introduction of an evaluation method and a novel approach to address compositional generalization fills a gap in the current research.
Reasons for Rejection:

The study is limited to a single model size, leaving questions about the method's efficacy in bridging compositional generalization gaps in larger models.
The proposed method seems prone to repetition issues, particularly in WebNLG content that spans an average of 1 to 2 sentences, and effective solutions to this concern aren't elaborated upon.
The CG-Random performs quite well in many scenarios, raising questions about the necessity and effectiveness of the complex predicate composition process.